# Membrane-mimetic thermal proteome profiling (MM-TPP) toward mapping membrane protein–ligand dynamic interactions

Rupinder Singh Jandu[1], Ashim Bhattacharya[1], Frank Antony[1], Mohammed Al-Seragi[1], Hiroyuki Aoki[2], Mohan Babu[2], Franck Duong van Hoa[1]*

[1]Department of Biochemistry and Molecular Biology, Life Sciences Institute, University of British Columbia, Vancouver, Canada; [2]Department of Biochemistry, University of Regina, Regina, Canada

## eLife Assessment

This **valuable** study introduces the peptidisc-TPP approach as a promising solution to challenges in membrane proteomics, enabling thermal proteome profiling in a detergent-free system. The concept is innovative and holds significant potential, and the demonstration of its utility and validation is **solid**. The method presents a strong foundation for broader applications in identifying physiologically and pharmacologically relevant membrane protein-ligand interactions.

*For correspondence:
fduong@mail.ubc.ca

**Abstract** Integral membrane proteins (IMPs) are central targets for small-molecule therapeutics, yet robust, unbiased, and detergent-free approaches to assess their on- and off-target interactions remain limited. Previously, we introduced the Peptidisc membrane mimetic (MM) for water-soluble stabilization of the membrane proteome and interactome (Carlson et al., eLife, 2019). In this work, we combine the Peptidisc with thermal proteome profiling (TPP) to establish membrane-mimetic thermal proteome profiling (MM-TPP), a method that enables proteome-wide mapping of membrane protein–ligand interactions. Using a membrane protein library derived from mouse liver tissue, we detected the specific effects of ATP and orthovanadate on the thermal stability of ATP-binding cassette (ABC) transporters, as well as stability shifts driven by the hydrotropic effect of ATP and its by-products on G protein-coupled receptors (GPCRs). In contrast, detergent-based TPP (DB-TPP) with ATP–$VO_4$ failed to yield specific enrichment of ATP-binding proteins, underscoring the unique capacity of MM-TPP. To further validate the approach, we demonstrated the ability of MM-TPP to detect specific ligand-induced stabilization of cognate targets, exemplified by the selective thermal stabilization of the P2RY12 receptor by 2-methylthio-ADP. Together, these findings position MM-TPP as a robust platform for uncovering both on- and off-target effects of small molecules, providing insights into the druggable membrane proteome and its stability in consequence of changing dynamic ligands.

## Introduction

Integral membrane proteins (IMPs) are essential to numerous cellular functions, including signal transduction, cell–cell recognition, and the transport of nutrients and ions across cellular membranes (*Boulos et al., 2023*). Although comprising only 20–30% of the human genome, they represent nearly two-thirds of druggable targets due to their significant impact on cellular physiology and their exposure to

the cell surface. This prevalence highlights the critical nature of IMPs in pharmacology (*Bakheet and Doig, 2009*; *Fagerberg et al., 2010*; *Helbig et al., 2010*; *Santos et al., 2017*). However, studying the interactions between small-molecule drugs and membrane proteins poses unique challenges, primarily due to the low abundance and hydrophobic characteristics of these proteins, which complicate their characterization and analysis.

Proteomics-based technologies have become powerful tools for identifying protein–ligand interactions (*Aebersold and Mann, 2016*; *Li et al., 2016*). Among these, affinity purification mass spectrometry (AP-MS) uses chemical compounds covalently attached to an affinity matrix to capture target proteins, allowing for detailed characterization of their interactions (*Dunham et al., 2012*; *Kawatani and Osada, 2014*). Another approach, drug affinity responsive target stability (DARTS), leverages ligand-induced conformational changes detected through limited proteolysis, offering insights into the dynamic nature of protein–ligand interactions (*Lomenick et al., 2009*; *Pai et al., 2015*). In addition to AP-MS and DARTS, thermal proteome profiling (TPP) has emerged as a complementary approach for studying protein–ligand interactions (*Lambos et al., 2025*). Derived from the cellular thermal shift assay, TPP enables the detection of ligand-induced conformational changes by assessing protein stability under heat stress (*Molina et al., 2013*; *Savitski et al., 2014*). Its relatively straightforward implementation has made it a popular choice in drug screening campaigns. However, TPP typically relies on cytosolic extracts and often overlooks pharmaceutically relevant transmembrane proteins (*Franken et al., 2015*; *Mateus et al., 2020*; *Lambos et al., 2025*).

To expand these techniques to membrane proteome coverage, detergent-based solubilization has been widely used (*Huber et al., 2015*; *Kalxdorf et al., 2021*; *Reinhard et al., 2015*). However, even mild detergents can disrupt native protein structures or drug associations, leading to artifacts that complicate drug target identification (*Berlin et al., 2023*; *Yang et al., 2014*; *Ye et al., 2023*). While crucial for solubilizing membrane proteins and enabling structural and functional studies (*O'Malley et al., 2011*; *Allison et al., 2015*; *Beckner et al., 2020*), detergents remain, as we show here, largely incompatible with workflows like TPP due to mass spectrometry (MS) limitations (*Brough et al., 2024*; *Antony et al., 2025*).

To overcome the limitations associated with detergent use, a variety of MMs have been developed to maintain membrane proteins in a water-soluble, native-like state (*Denisov and Sligar, 2016*; *Dörr et al., 2014*; *Young, 2023*). Among these, the Peptidisc has emerged as a versatile, self-assembling scaffold characterized by its 'one-size-fits-all' property, capable of stabilizing IMPs of diverse sizes and topologies (*Carlson et al., 2018*). This adaptability has enabled the comprehensive isolation of membrane proteomes from both bacterial and mammalian sources, yielding the so-called Peptidisc libraries that are directly compatible with downstream MS workflows (*Antony et al., 2024*; *Carlson et al., 2019*; *Zhao et al., 2023*). In addition to MS compatibility, these libraries also capture and stabilize IMPs in their functional states while preserving their interactomes and lipid allosteric modulators,

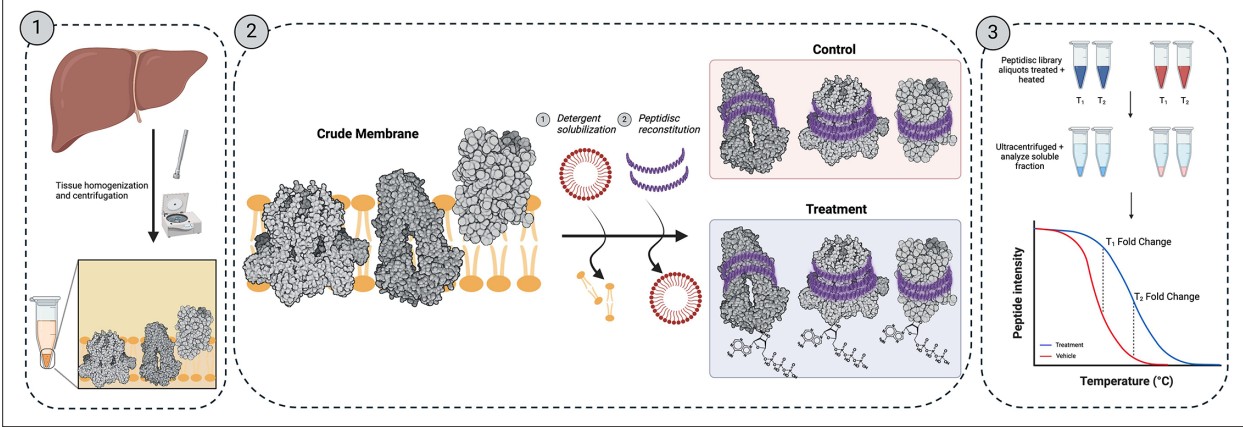

**Figure 1.** The membrane-mimetic thermal proteome profiling (MM-TPP) experimental workflow. (**1**) Crude membranes are prepared from the liver organ. (**2**) Integral membrane proteins (IMPs) are solubilized with detergent and reconstituted in the Peptidisc library. The water-soluble library is exposed to the ligand of interest (treatment) or the corresponding vehicle (control). (**3**) Protein samples are heated at specific temperatures to induce precipitation, followed by ultracentrifugation. The soluble fraction is analyzed by mass spectrometry to detect changes in protein abundances between the treatment and control samples.

thus enabling a more comprehensive analysis of membrane protein dynamics and ligand interactions (*Angiulli et al., 2020*; *Carlson et al., 2019*; *Jandu et al., 2024*; *Urner et al., 2022*).

Building on this foundation, we thought to integrate the Peptidisc into the TPP workflow, introducing a straightforward and detergent-free approach termed membrane-mimetic TPP (MM-TPP). Here, we apply the MM-TPP workflow to bacterial and mouse liver membrane proteomes, demonstrating its ability to detect ATP-binding proteins, including ATP-binding cassette (ABC) transporters, G protein-coupled receptors (GPCRs), and multi-subunit complexes. Notably, this approach is sensitive enough to capture interactions mediated by ATP by-products, offering critical insights into both system-wide and ligand-specific dynamics—factors that are essential to consider in small-molecule drug development.

## Results and discussion

The MM-TPP workflow is illustrated in *Figure 1*. Initially, the detergent-solubilized membrane fraction is reconstituted into Peptidisc libraries, as described in previous studies (*Antony et al., 2024*; *Carlson et al., 2019*; *Zhao et al., 2023*). The library is subsequently divided into two aliquots: one exposed to the ligand of interest (treatment) and the other treated with ddH$_2$O (control). Afterward, the samples are heated for 3 min to facilitate protein denaturation and precipitation. The soluble fraction is then isolated via ultracentrifugation and analyzed by liquid chromatography–tandem mass spectrometry (LC–MS/MS). Proteins exhibiting significant stabilization or destabilization are identified using the methodology outlined by *Zhang et al., 2020*, in which proteins that meet defined fold difference thresholds between triplicate treatment and control groups are considered highly probable ligand binders.

As a validation step for the MM-TPP approach, we determined the thermal stability of the purified and Peptidisc-reconstituted bacterial ABC transporter MsbA in the presence of ATP and vanadate (VO$_4$). Vanadate is a potent inhibitor of the ABC transporter family and, upon binding with ATP, stabilizes the transporter in a specific conformation (*Lyu et al., 2022*). Consistent with this mechanism, SDS–PAGE analysis revealed increased thermal stability of MsbA in the presence of these ligands (*Figure 2A*, *Figure 2—figure supplement 1*). To extend this observation to a more native context, we further assessed the effect of ATP–VO$_4$ on a Peptidisc library derived from wild-type *Escherichia coli*, where MsbA is present at endogenous expression levels (*Figure 2B*). In the control sample, MsbA peptide abundance progressively decreased at 51°C, 56°C, and 61°C relative to the bulk proteome. In contrast, in the presence of ATP-VO$_4$, MsbA was significantly stabilized at 51°C and above (*Figure 2C*), demonstrating that our approach can detect thermal stabilization even in proteins expressed at low endogenous levels. Notably, at 61°C, the volcano plot revealed that MsbA stabilization coincided with that of two other IMPs, FtsK and LolC (*Figure 2D*). FtsK, one of the longest membrane proteins in *E. coli* (1329 amino acids), belongs to the AAA ATPase family, while LolC is the membrane transport domain of the soluble ABC subunit LolD (*Sharma et al., 2021*). In contrast, DgkA, which is a small diacylglycerol kinase comprising 122 amino acids, exhibited destabilization under the same conditions (*Figure 2D*; *Li et al., 2015*; *Zheng and Jia, 2013*). Collectively, these results validate MM-TPP as a robust and straightforward method for identifying ATP binders within complex mixtures of soluble proteins and IMPs.

We next applied the MM-TPP workflow to mouse liver tissue, where drug screening for cell surface IMPs is particularly relevant. As expected, the total number of proteins identified decreased following heat treatment (*Figure 3A*). Notably, in this detergent-free environment, this reduction affected soluble and membrane proteins equally, with the proportion of IMPs remaining stable across temperatures—approximately 48% (*Supplementary file 2*). We then performed MM-TPP in the presence of ATP–VO$_4$ (*Figure 3B*). Consistent with results from the *E. coli* library, most proteins exhibited increased thermal stability upon ATP-VO$_4$ treatment, as indicated by the rightward shift of data points in the plot. Recent studies suggest that ATP can function as a natural hydrotrope, influencing the global stability of protein populations by interacting with thermally sensitive regions (*Ou et al., 2021*; *Patel et al., 2017*). Important to this study and as expected, Gene Ontology (GO) term analysis of significantly stabilized IMPs revealed enrichment in functions related to nucleoside-phosphate binding and primary active transport (*Figure 3C*). Supporting this, thermal stability profiles across different temperatures confirmed a marked sensitivity of ABC transporters. Among the ten ABC transporters identified in the liver library, eight showed significant thermal stabilization in the presence of

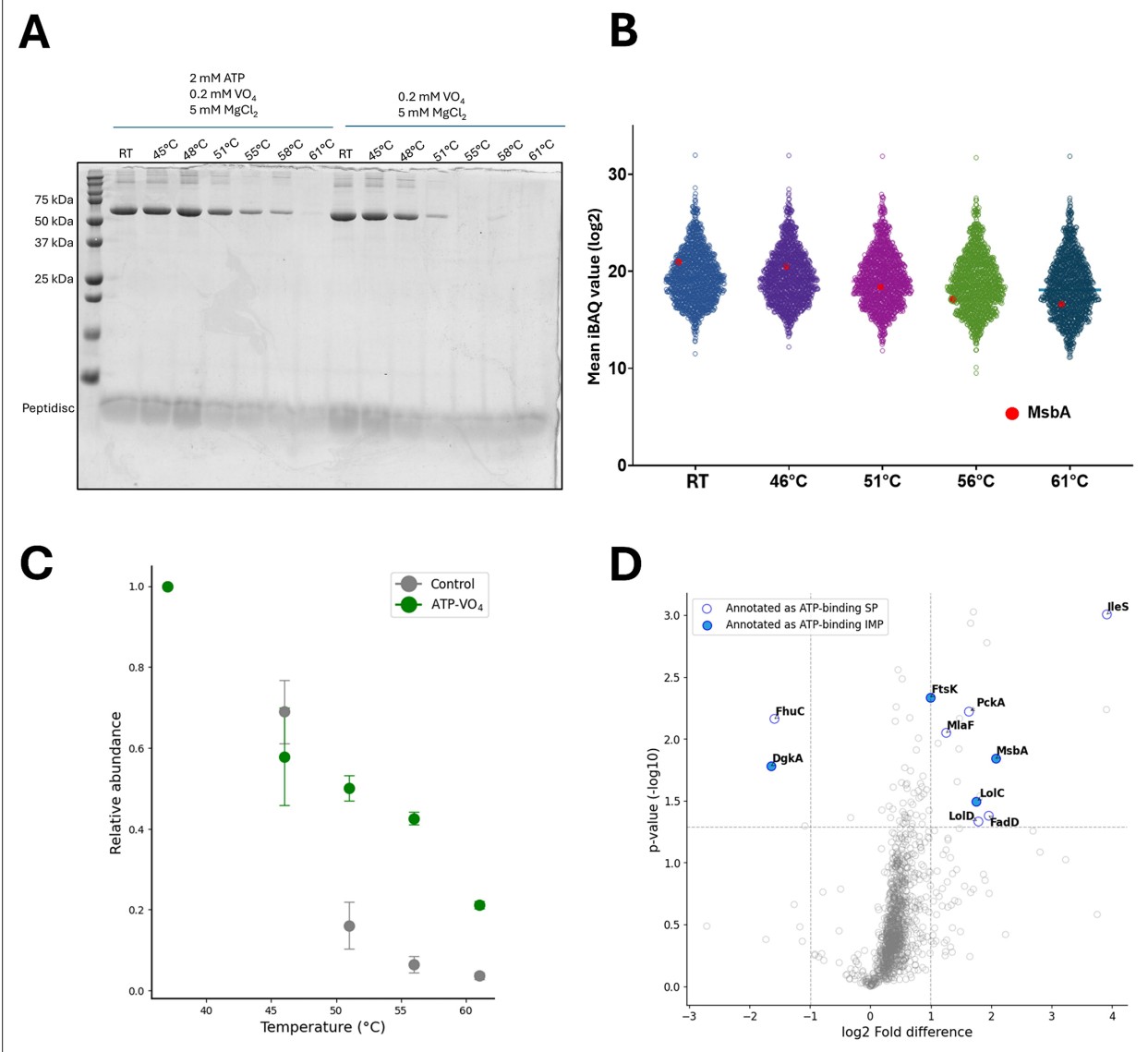

**Figure 2.** Membrane-mimetic thermal proteome profiling (MM-TPP) of integral membrane proteins (IMPs) prepared from *E. coli*. (**A**) Stability of purified MsbA in Peptidisc in the presence of the indicated ligands. Samples are heat-treated and centrifuged before analysis by 12% SDS–PAGE and Coomassie blue staining. (**B**) Grouped scatterplot representation of mean log$_2$-transformed IBAQ value obtained for all identified proteins in the *E. coli* library at the indicated temperatures. The location of MsbA on the plot is shown as a red dot. The mean value is obtained from three replicates of the temperature exposure assay (n=3). (**C**) Relative abundance of MsbA based on the label-free quantification (LFQ) peptide intensities obtained across temperatures in the presence of ATP + orthovanadate (ATP–VO$_4$; orange) compared to a control sample (gray). Data is a mean ± standard deviation from triplicate assays (n=3). (**D**) Volcano plot analysis of stabilized and destabilized proteins following ATP–VO$_4$ exposure at 61°C based on log$_2$-transformed LFQ peptide intensities. A log$_2$ fold difference significance cutoff of +1 and –1 with a –log$_{10}$ p-value cutoff of p>1.3 is applied. Hollow blue dots indicate annotated ATP-binding soluble proteins (SP), and solid blue dots indicate ATP-binding IMPs. Data represent the mean from three replicates (n=3).

The online version of this article includes the following source data and figure supplement(s) for figure 2:

**Source data 1.** Original gel images for *Figure 2* and *Figure 2—figure supplement 1*.

**Source data 2.** Original gel images with annotations for *Figure 2* and *Figure 2—figure supplement 1*.

**Figure supplement 1.** Thermal stability of MsbA in detergent and in Peptidisc with different ligands.

ATP–VO$_4$ at one or more temperature points (*Table 1* and *Figure 3D*). Besides ABC transporters, the BCS1L protein exhibited a remarkable ~30-fold increase in stability, which was further enhanced by AMP-PNP (*Figure 3E*, *Figure 3—figure supplement 1*). This enhanced stabilization is consistent with recent structural studies demonstrating substantial conformational changes in the heptameric BCS1L

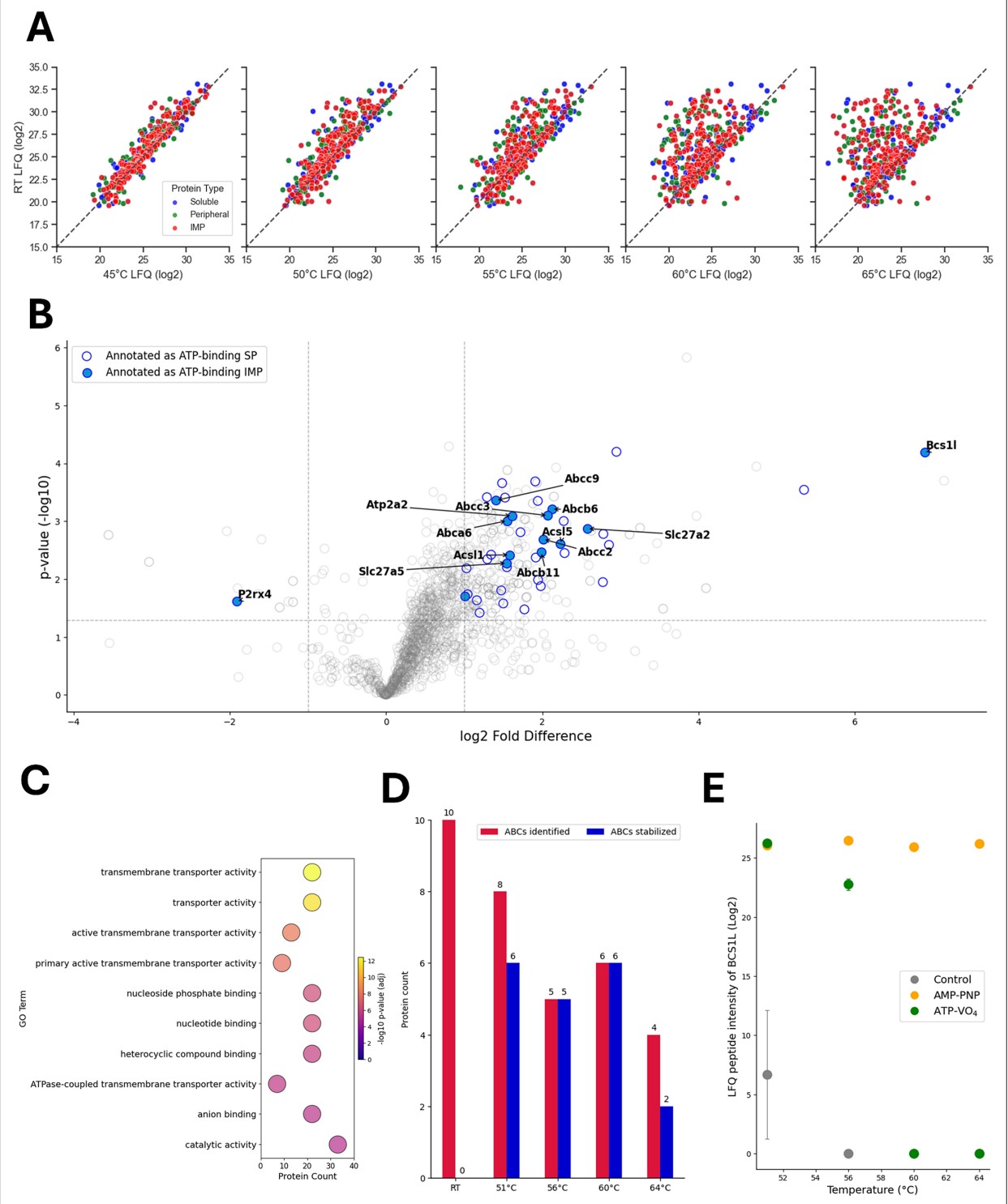

**Figure 3.** Membrane-mimetic thermal proteome profiling (MM-TPP) of integral membrane proteins (IMPs) prepared from the mouse liver tissue. (**A**) Global protein intensities derived from log$_2$-transformed label-free quantification (LFQ) values of peptidisc-reconstituted liver extract. The plot displays soluble proteins (blue), membrane-associated (green), and IMPs (red) identified at the indicated temperatures compared to room temperature (RT). The dashed line is the identical value line. (**B**) Volcano plot of stabilized and destabilized proteins based on log$_2$-transformed LFQ values at 51°C based on a fold difference cutoff of >1 or <–1 and –log$_{10}$ p-value of >1.3. The soluble proteins (SP) annotated as ATP binding are represented as hollow blue circles, and the IMPs annotated as ATP binding are represented as solid blue circles. The mean value is obtained from three replicates at the temperature exposure assay (n=3). (**C**) Gene Ontology (GO) term enrichment analysis of molecular functions of stabilized IMPs identified in B. The presented top 10 significant terms are based on adjusted p-value (false discovery rate [FDR] = 5% after Benjamini-Hochberg correction). (**D**) Number

*Figure 3 continued on next page*

*Figure 3 continued*

of ATP-binding cassette (ABC) transporters identified and stabilized by ATP–VO$_4$ at the indicated temperatures. (**E**) Log$_2$-transformed LFQ peptide intensities of BCS1L in the presence of ATP–VO$_4$ (green), AMP-PNP (orange), and vehicle control (gray) across the temperature range. Data is a mean ± standard deviation from three replicates (n=3).

The online version of this article includes the following figure supplement(s) for figure 3:

**Figure supplement 1.** Volcano plot of stabilized and destabilized proteins at 51°C in the presence of AMP-PNP.

complex upon binding of this non-hydrolyzable ATP analog (*Pan et al., 2023*; *Tang et al., 2020*; *Zhan et al., 2024*).

To further assess the effectiveness of MM-TPP, we directly compared its performance to a detergent-based TPP assay (DB-TPP) at 51°C. In both workflows, crude membrane fractions from mouse liver were solubilized in 1% DDM to extract membrane proteins. However, while MM-TPP involves reconstitution of solubilized proteins into Peptidiscs before heat treatment, the DB-TPP workflow omits this step, keeping the proteins in detergent solution throughout. Following thermal treatment, DB-TPP also includes an additional detergent removal step for MS analysis. This side-by-side comparison revealed key differences in the TTP profiles. Overall, DB-TPP provided broader proteome coverage, including a greater number of soluble proteins identified (*Table 2*). In contrast, in the MM-TPP workflow, soluble proteins were actively depleted during Peptidisc reconstitution, resulting in an IMP-enriched proteome. Crucially, at 51°C, 7.4% of all identified proteins and 6.4% of differentially stabilized proteins in the DB-TPP dataset were annotated as ATP binding, indicating that the apparent differential stabilization is likely stochastic rather than driven by specific ligand binding (*Figure 4A*). By comparison, in MM-TPP, 9.3% of all identified proteins and 17% of differentially stabilized proteins were annotated as ATP binding, demonstrating selective enrichment for ATP–VO$_4$-mediated stabilization events (*Figure 3B*).

Beyond demonstrating improved selectivity for ATP–VO$_4$-mediated stabilization, we next asked whether MM-TPP could also resolve more selective ligand–protein interactions. To test this, we applied the method using 2-methylthio-ADP (2-MeS-ADP), a well-characterized agonist of the P2RY12 receptor (*Zhang et al., 2014*). MM-TPP revealed clear and reproducible thermal stabilization of P2RY12, showing more than a sixfold increase in stability at both 51°C and 57°C (–log$_{10}$p>5.97; *Figure 4B* and *Figure 4—figure supplement 1*). Notably, no other proteins, including the structurally related but nonresponsive P2RY6 receptor, exhibited comparable stabilization, underscoring MM-TPP's ability to detect highly specific ligand–receptor interactions amidst a complex membrane proteome.

Next to these highly specific receptor–ligand interactions, we also observed a broader spectrum of thermal stabilization events that lacked direct annotation to ATP binding. For instance, a substantial proportion of IMPs without known ATP-binding function was still significantly stabilized in the presence of ATP–VO$_4$. For example, among the 178 IMPs that passed the significance threshold, approximately

**Table 1.** ATP-binding cassette transporters detected in the mouse liver Peptidisc library.
A minimum of two unique peptides was required to identify the protein at a given temperature. The exact number of unique peptides identified for each protein under each ligand and temperature condition is provided in *Supplementary file 2*. Stabilization was defined using a log$_2$ fold change > 1 between treatment and control samples, with a –log10 p>1.3, calculated from replicate samples (n = 3).

| Uniprot-ID | Protein name | Full protein name | Stabilized temperatures (°C) |
|---|---|---|---|
| Q8K441 | ABCA6 | ATP-binding cassette subfamily A member 6 | 51, 56, 60, 64 |
| J3QNY6 | ABCB11 | ATP-binding cassette, subfamily B (MDR/TAP), member 11 | 51, 56, 60 |
| Q9DC29 | ABCB6 | ATP-binding cassette subfamily B (MDR/TAP), member 6 | 51 |
| Q8VI47 | ABCC2 | ATP-binding cassette subfamily C (CFTR/MRP), member 2 | 51, 56, 60 |
| A0A0R4J015 | ABCC3 | ABC-type glutathione-*S*-conjugate transporter (CFTR/MRP) | 51, 60 |
| P70170 | ABCC9 | ATP-binding cassette subfamily C member 9 (Sulfonylurea receptor 2) | 51, 56, 60 |
| S4R2E1 | ABCG2 | ATP-binding cassette subfamily G member 2 (Urate exporter) | 56, 60, 64 |
| Q99PE8 | ABCG5 | ATP-binding cassette subfamily G member 5 (Sterolin-1) | 51 |

**Table 2.** Comparison of ATP-binding protein stabilization in membrane-mimetic thermal proteome profiling (MM-TPP) and detergent-based thermal proteome profiling (DB-TPP).

Reported values at each temperature represent the total number of proteins meeting the inclusion criteria (at least two unique peptides) across triplicate control and treatment conditions. As protein counts are determined post-analysis, they are reported as single values rather than as means with standard deviations. Each protein was identified based on at least two unique peptides (n = 3).

| | MM-TPP | | | | DB-TPP | | | |
| | | | SPs | IMPs | | | SPs | IMPs |
| Temperature (°C) | Total protein | IMPs | % ATP binders stabilized | % ATP binders stabilized | Total protein | IMPs | % ATP binders stabilized | % ATP binders stabilized |
|---|---|---|---|---|---|---|---|---|
| 51 | 1380 | 419 (30%) | 33.7% | 37.9% | 1862 | 428 (23%) | 1.74% | 0% |
| 56 | 1179 | 369 (31%) | 39.2% | 45.6% | 1839 | 428 (23%) | 0% | 0% |
| 60 | 1090 | 344 (32%) | 22.9% | 48.4% | 1621 | 391 (24%) | 0% | 0% |
| 64 | 992 | 316 (32%) | 47.0% | 58.1% | 1539 | 372 (24%) | 0% | 5.56% |

43% were annotated with GO terms related to nucleic acid/nucleotide binding, GTP binding, phosphate binding, and other related functions (*Figure 5A*). Some of these IMPs may achieve stabilization indirectly through interactions with ATP-binding partners, as illustrated by the LolC/LolD complex reported above (*Figure 2D*). Others may exhibit off-target stabilization effects, potentially due to ATP metabolites such as ADP, AMP, or inorganic phosphate, which are known to influence protein conformation and stability. This scenario is exemplified by the purinergic receptor P2RY6, a GPCR that responds preferentially to ADP rather than ATP (*von Kügelgen and Hoffmann, 2016*), which showed the highest differential thermal stabilization among all IMPs at 56°C (*Supplementary file 1*). Similarly, P2RY12, a member of the same GPCR family, showed significant stabilization across 56°C, 60°C, and 64°C (*Supplementary file 1*), consistent with its known preference for ADP (*Entsie et al., 2023*; *Zhang et al., 2014*). Supporting this interpretation, neither receptor exhibited thermal stabilization in the presence of the non-hydrolyzable ATP analog AMP-PNP (*Figure 5B*). Conversely, the trimeric ATP-gated cation channel P2RX4 showed significant destabilization in response to ATP–$VO_4$ at 51°C and 64°C (*Figure 3B*) and AMP-PNP at 56°C and 64°C (*Figure 5C*), consistent with previous reports highlighting the unique ATP sensitivity of P2RX family members compared to the P2RY family (*Carnero Corrales et al., 2021*; *Thompson et al., 2021*).

Among all observed side effects, the most striking stabilization was detected for the FAD-containing monoamine oxidase B (Mao-B) at 64°C (*Figure 5—figure supplement 1*). To explore the potential basis for this off-target effect, we used AlphaFold3 to model the binding of ATP and its derivatives within the FAD-binding pocket of Mao-B (*Abramson et al., 2024*). Interestingly, the best-fitting models were obtained for ADP and AMP (pLDDT >90), while ATP yielded the lowest confidence score (*Figure 5D*). Although this computational result remains hypothetical, it offers a plausible structural rationale that warrants further experimental validation.

Finally, to validate the robustness of these findings, we assessed the reproducibility of MM-TPP by repeating the ATP–$VO_4$ experiment at 51°C using data-independent acquisition (DIA) (*Figure 5—figure supplement 2*), in contrast to the data-dependent acquisition (DDA) employed initially. The results were highly consistent with the previous dataset. Overall, 7.8% of all identified proteins were annotated as ATP binding, with this proportion rising to 17% among proteins exhibiting a $\log_2$ fold change greater than 0.5—mirroring the enrichment observed in the DDA-based analysis. Notably, BCS1L and SLC27A2 displayed strong thermal stabilization ($\log_2$ fold change >1), while several ABC transporters, including ABCB11, ABCG2, ABCD3, ABCG8, and ABCA1, showed intermediate stabilization ($\log_2$ fold changes between 0.5 and 1). In line with earlier results, P2RX4 was again significantly destabilized ($\log_2$ fold change < –1). These findings confirm that MM-TPP produces consistent and

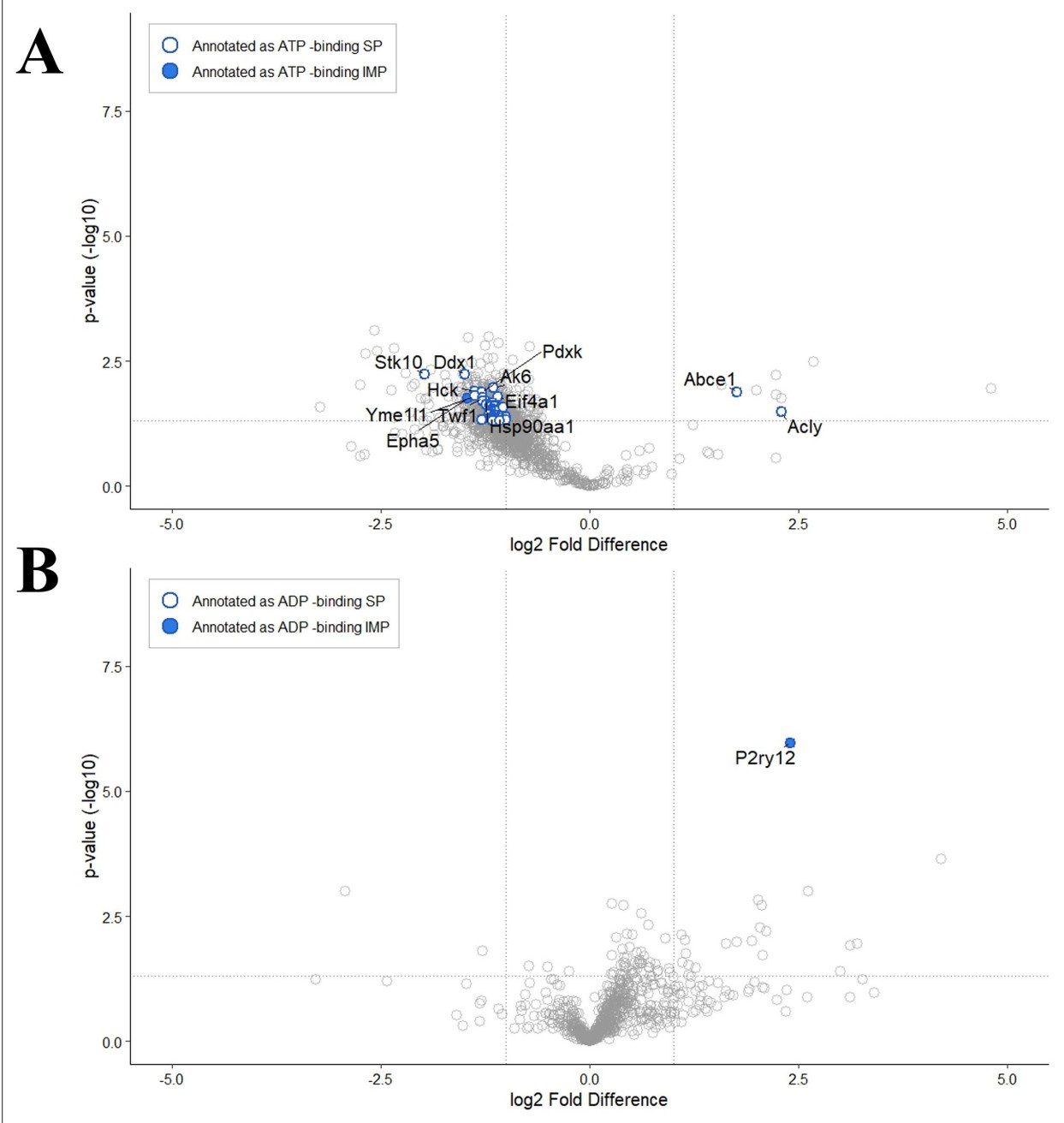

**Figure 4.** Protein stabilization profiles at 51°C with ATP–VO$_4$ in detergent-based thermal proteome profiling (DB-TPP) and 2-MeS-ADP in membrane-mimetic thermal proteome profiling (MM-TPP). (**A**) Volcano plot of stabilized and destabilized proteins at 51°C in detergent-based TPP (DB-TPP) with ATP–VO$_4$. Log$_2$-transformed label-free quantification (LFQ) peptide values were used, with thresholds set at a fold change>1 or <–1 and a –log$_{10}$ p-value>1.3. Soluble proteins (SP) annotated as ATP binding are shown as hollow blue circles, and integral membrane proteins (IMPs) annotated as ADP binding are shown as solid blue circles. Data represent the mean of three biological replicates (n=3). (**B**) Volcano plot of stabilized and destabilized proteins at 51°C in MM-TPP with 2-methylthio-ADP (2-MeS-ADP), displayed using the same thresholds and annotations as in panel **A**.

The online version of this article includes the following figure supplement(s) for figure 4:

**Figure supplement 1.** Volcano plot of stabilized and destabilized proteins at 57°C in the presence of 2-MeS-ADP.

reproducible profiles of protein stabilization and destabilization across distinct MS acquisition strategies, further supporting its utility for ligand engagement studies in complex membrane proteomes.

Together, these results validate the integration of Peptidisc libraries into TPP for broad-scale detection of membrane protein–ligand interactions. MM-TPP not only enhances the recovery and stability

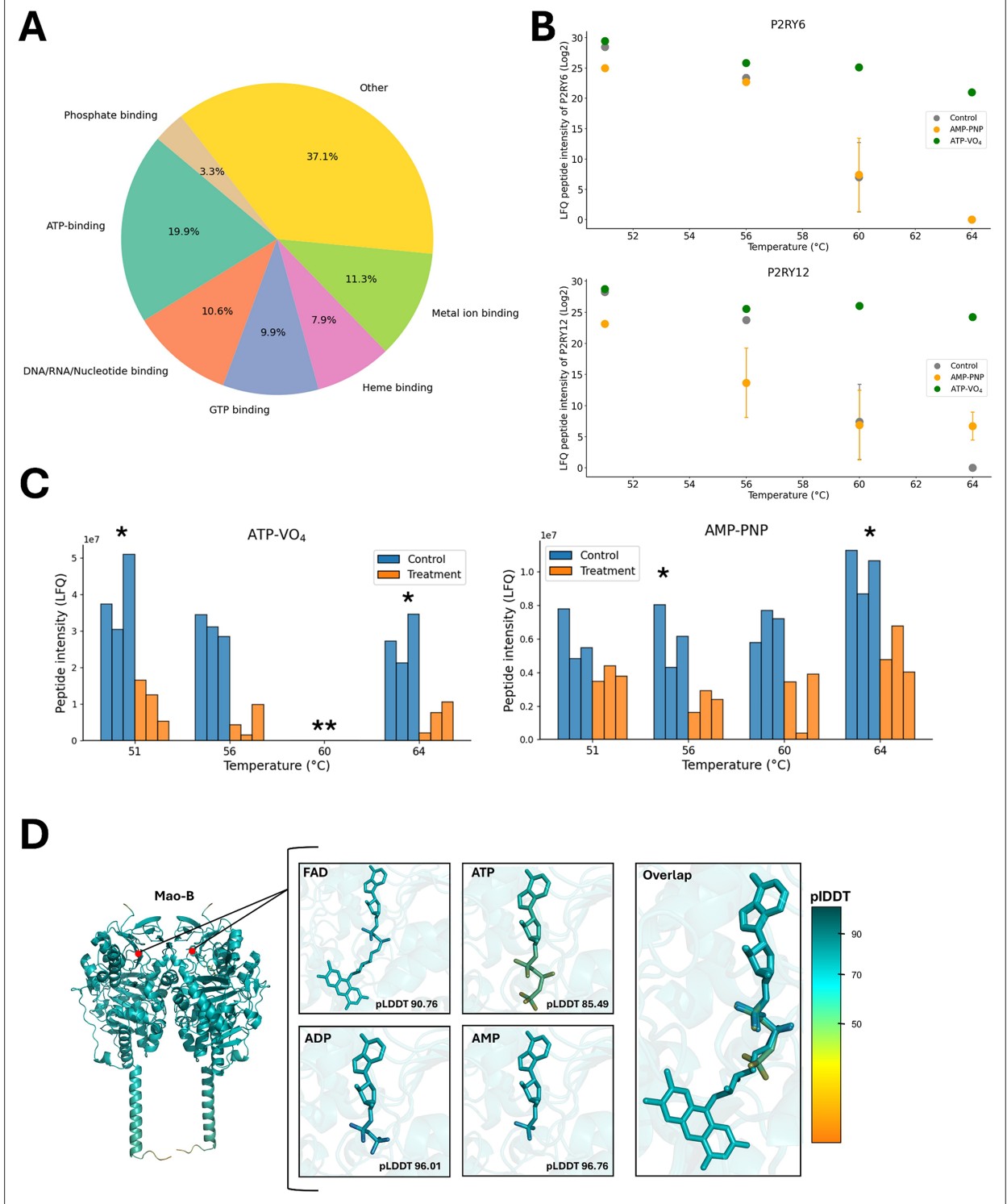

**Figure 5.** ATP and analog-induced off-target stabilization of liver membrane proteins. (**A**) Gene Ontology (GO) term analysis of molecular functions and distribution for all integral membrane proteins (IMPs) (n=178) significantly stabilized with ATP–VO$_4$ at temperatures tested with the mouse liver library. (**B**) Log$_2$-transformed label-free quantification (LFQ) peptide intensity variations of P2RY6 and P2RY12 over the temperature range with AMP-PNP (orange), ATP–VO$_4$ (green), or none (gray). Data is a mean ± standard deviation from three replicates (n=3). (**C**) Relative LFQ peptide intensity variations of P2RX4 at the indicated temperature in the presence of ATP–VO$_4$ (left panel) or AMP-PNP (right panel). Data from treatment samples (orange) and control samples (blue) is from triplicates (n=3). * Represents p-value≤0.05. **Protein not detected. (**D**) Structural model of homodimeric Mao-B with the predicted binding of FAD, ATP, ADP, and AMP ligands within the FAD-binding pocket, as indicated as red dots. Each ligand is presented individually in the FAD-binding pocket or as an all-ligand overlap generated by *AlphaFold3*. The respective predicted local distance difference test (pLDDT) score for

*Figure 5 continued on next page*

*Figure 5 continued*

each ligand is shown, with higher scores representing more favorable ligand fitting. The color gradient represents a high pLDDT score as blue and a low pLDDT score as orange.

The online version of this article includes the following figure supplement(s) for figure 5:

**Figure supplement 1.** Volcano plot of stabilized and destabilized proteins at 64°C in the presence of ATP–VO$_4$.

**Figure supplement 2.** Correlation plot of protein differential stabilization at 51°C in membrane-mimetic thermal proteome profiling (MM-TPP), performed with data-independent acquisition (DIA) from a whole-mouse liver library.

of IMPs but also enables direct profiling of membrane proteomes from native tissues and organs. This preserves critical biological features, such as endogenous protein expression levels, native protein–protein interactions, posttranslational modifications, and tissue-specific isoforms, that are often disrupted in traditional cell culture systems (*Kwasnik et al., 2016*; *Perez-Perri et al., 2023*).

Although the Peptidisc workflow does not bypass the need for initial detergent solubilization, it eliminates key downstream limitations associated with detergent-based methods. In particular, MM-TPP preserves ligand binding and is fully compatible with MS, unlike many detergent-based workflows where residual detergent can interfere with both ligand interactions and MS sensitivity (*Brough et al., 2024*; *Lambos et al., 2025*). As such, MM-TPP is inherently complementary to detergent-based approaches while offering clear advantages for downstream proteomic analyses. To further expand its applicability, future studies could explore a broader range of nucleotide analogs, including antagonists such as TNP-ATP or ATPγS, and clinically relevant small molecules, such as the CFTR modulator VX-809. These applications would further underscore MM-TPP's pharmacological potential in profiling membrane protein function and drug interactions.

While challenges remain, such as maintaining physiological signaling contexts, integrating MMs like Peptidiscs early in drug discovery offers a powerful strategy. MM-TPP captures both direct target engagement and off-target effects from parent compounds or their metabolites, providing functional insights into ligand dynamics and specificity, which are important considerations in early-stage drug development and safety profiling.

## Methods

### Preparation of *E. coli* crude membranes

Wild-type *E. coli* BL21 (DE3) was grown in 1 L of LB medium. After 3 hr, cells were harvested by low-speed centrifugation (6000×*g*, 6 min) and resuspended in Buffer A (50 mM Tris–HCl pH 7.8, 100 mM NaCl, 10% glycerol) supplemented with 1 mM phenylmethylsulfonyl fluoride (1 mM). Cells were lysed through a microfluidizer (Microfluidics; 3 passes at 15,000 psi at 4°C). Unbroken cells and large aggregates were removed by low-speed centrifugation (6000×*g*, 6 min). The crude membrane fraction was isolated by ultracentrifugation (100,000×*g*, 45 min, 4°C, Beckman Coulter rotor Ti70). Membranes were resuspended in Buffer A at 5 mg/mL and stored at –80°C for later use.

### Expression and purification of His-MsbA for thermal stability assay

Histidine-tagged MsbA (His-MsbA) was expressed, purified, and reconstituted in Peptidisc through the on-bead method as described in *Angiulli et al., 2020*, with slight modifications (*Angiulli et al., 2020*). Briefly, His-MsbA was produced in *E. coli* BL21(DE3) at 37°C in 1 L of LB medium supplemented with 50 µg/mL kanamycin. The inducer IPTG (0.5 mM) was added during the exponential growth phase (OD$_{600nm}$ ~ 0.4). After 3 hr, cells were harvested by low-speed centrifugation (6000×*g*, 6 min) and resuspended in Buffer A (50 mM Tris–HCl pH 7.8, 100 mM NaCl, 10% glycerol) supplemented with phenylmethylsulfonyl fluoride (1 mM). Cells were lysed through a microfluidizer (Microfluidics; 3 passes at 15,000 psi at 4°C). Unbroken cells and large aggregates were removed by low-speed centrifugation (6000×*g*, 6 min). The crude membrane fraction containing MsbA was isolated by ultracentrifugation (100,000×*g*, 45 min, 4°C, Beckman Coulter rotor Ti70). About 2 mg of MsbA-enriched membranes were solubilized with 1% DDM (wt/vol) for 30 min at 4°C. The detergent-solubilized material was ultracentrifuged (180,000×*g*, 15 min, 4°C, Beckman Coulter rotor TLA55) to pellet insoluble material. The supernatant was incubated with Ni-NTA (150 µL) resin for 45 min at 4°C on a tabletop rocker. The resin was sedimented through centrifugation (2000×*g*, 1 min) and washed three times with

1.5 mL of Buffer A supplemented with 0.02% DDM and 30 mM imidazole. After removing the excess buffer, the beads were resuspended in Buffer A containing Peptidisc peptide (1 mg/mL). The excess peptide was washed away with Buffer A. The protein was eluted with 600 mM imidazole in Buffer A, and the concentration was determined through a Bradford reagent.

## Thermal stability assay of His-MsbA with ATP–VO$_4$

Purified His-MsbA (0.5 mg/mL) in Buffer A supplemented with 5 mM MgCl$_2$ and 0.2 mM orthovanadate (VO$_4$) was separated into two equal volumes designated as treatment and control samples. To the treatment sample, ATP disodium trihydrate was added to a final concentration of 2 mM, and an equal volume of ddH$_2$O was administered to the control. Samples were incubated at room temperature for 10 min prior to aliquoting 50 µL into separate 0.2 µL PCR tubes. These aliquots were subjected to temperatures ranging from 45°C to 61°C for 3 min, followed by ultracentrifugation at 180,000×$g$ for 15 min at 4°C to pellet the unfolded proteins. The supernatants were collected, and 10 µL from each sample was analyzed on a 12% SDS–PAGE.

## Harvest of *Mus musculus* liver

The C57BL/6 mice were kept in specific pathogen-free conditions and received humane care in compliance with the Canadian Council of Animal Care guidelines, and the animal protocol A23-0280 was approved by the Animal Care Committee of the University of British Columbia. The mouse organs were obtained from female mice. The authors acknowledge that they did not consider the impact of mouse sex at the time of the study design. The mice received standard chow. At the age of 12 weeks, the mice were sacrificed, and the liver was excised and placed in vials with ice-cold sterile phosphate-buffered saline (PBS) until processing.

## *M. musculus* liver processing and preparation of crude membranes

The excised organs were washed several times in ice-cold PBS to remove the blood and then minced and homogenized in ice-cold hypotonic lysis buffer (10 mM Tris–HCl pH 7.4, 30 mM NaCl, and 1 mM EDTA, 1× cocktail protease inhibitor, and 1 mM PMSF) using a tight-fit metal douncer. All subsequent steps were performed at 4°C or on ice. After adding 10 mM MgCl$_2$ and 50 µg/mL DNase, the suspension was further dounced and incubated for 10 min. The swollen tissue suspension was lysed using a French press (3 passes at 500 PSI). Unbroken cells and nucleus fraction were removed by low-speed centrifugation (1200×$g$ for 10 min). The supernatant was collected and centrifuged at a higher speed (5000×$g$, 10 min) to remove the mitochondrial fraction. The crude membrane fraction was then collected by ultracentrifugation (110,000×$g$, 45 min) in a Beckman TLA110 rotor. This membrane pellet was resuspended in 100 µL of 50 mM Tris, pH 7.9, 100 mM NaCl, and 10% glycerol (TSG Buffer) and stored at −80°C until use.

## Preparation of Peptidisc membrane protein library

For Peptidisc library preparation, ~2 mg of crude membranes (either *E. coli* or *M. musculus*) was solubilized in 50 mM Tris, pH 7.9, 25 mM NaCl, and 1% DDM (Solubilization Buffer) for 30 min at 4°C with gentle shaking. The insoluble material was pelleted by ultracentrifugation (180,000×$g$, 15 min). The detergent extract (500 µL) was then reconstituted by mixing it with a threefold excess (wt/wt) Peptidisc peptide for 15 min at 4°C. The sample was rapidly diluted to 15 mL in 50 mM Tris, pH 7.9 and 25 mM NaCl (Buffer A) over a 100 kDa cutoff centrifugal filter. The sample was concentrated (3000×$g$, 10 min) to ~200 µL. This process was repeated for a total of three rounds of dilution and concentration to deplete DDM to an approximate concentration of 0.008% to complete Peptidisc reconstitution. The Peptidisc library was immediately used for downstream TPP.

## MM-TPP profiling on membrane proteomes

Peptidisc membrane protein library was split into two equal aliquots representing the control and treatment samples. Both the treatment and control groups were supplemented with 5 mM MgCl$_2$ and 0.2 mM VO$_4$. Treatment samples were then exposed to a final concentration of 2 mM ATP, while the control received an equal volume of ddH$_2$O. For experiments with AMP-PNP, 2 mM AMP-PNP was substituted in place of ATP and VO$_4$ was omitted. For experiments with 2-Mes-ADP, 0.5 mM 2-Mes-ADP was substituted in place of ATP and VO$_4$ was omitted. The samples were incubated for 10 min at

room temperature, divided into four aliquots, and transferred into 0.2 mL PCR tubes. Each sample was heated in parallel for 3 min to its respective temperature (51–64°C), as previous studies have indicated that a 3–5 min exposure is optimal for visualizing protein denaturation (*Molina et al., 2013*; *Mateus et al., 2020*). Subsequently, the samples were centrifuged at 180,000×$g$ for 15 min at 4°C, and the supernatant was collected for in-solution digestion.

## DB-TPP on membrane proteomes

For DB-TPP, ~2 mg of crude membranes from *M. musculus* was solubilized in Solubilization Buffer for 30 min at 4°C with gentle shaking. The insoluble material was pelleted by ultracentrifugation (180,000×$g$, 15 min). The detergent-solubilized library was split into two equal aliquots representing the control and treatment samples. Both the treatment and control groups were supplemented with 5 mM MgCl$_2$ and 0.2 mM VO$_4$. Treatment samples were then exposed to a final concentration of 2 mM ATP, while the control received an equal volume of ddH$_2$O. The samples were incubated for 10 min at room temperature, divided into four aliquots, and transferred into 0.2 mL PCR tubes. Each sample was heated in parallel for 3 min to its respective temperature (51–64°C). Subsequently, the samples were centrifuged at 180,000×$g$ for 15 min at 4°C, and the supernatant was collected for processing as described by *Johnston et al., 2022*. Briefly, silica beads (9–13 μm diameter) were resuspended in water, washed once with 100% acetonitrile (CAN), rinsed twice with water, and resuspended at 50 mg/mL final. After each wash, the beads were isolated by brief centrifugation at 16,000×$g$ for 1 min. Crude membranes (~1 mg) were resuspended in ice-cold TS buffer (50 mM Tris–HCl, pH 8.0, 100 mM NaCl) supplemented with 1% (wt/vol) SDC for 30 min at 4°C with gentle shaking. The detergent extract was clarified by ultracentrifugation (110,000×$g$ for 15 min at 4°C), and aliquots (100 μg each) were gently vortex-mixed with glass beads (1 mg). CAN was then added to a final concentration of 80%, and samples were centrifuged at 16,000×$g$ for 5 min. The beads were rinsed three times with 500 μL of 80% ethanol without disturbing the pellet. After a final wash, the beads underwent MS sample preparation and LC–MS/MS analysis.

## MS sample preparation and LC–MS/MS analysis

Equal volumes of supernatants from the treatment and control groups were treated with 6 M urea at room temperature for 30 min before reduction with 10 mM fresh dithiothreitol (DTT) for 1 hr. Alkylation was performed with 20 mM iodoacetamide in the dark at room temperature for 30 min, followed by a second round of reduction via 10 mM DTT for 30 min. The urea was diluted to 1 M with Buffer A. Trypsin digestion was performed with an enzyme/protein ratio of 1:100 at room temperature for 24 hr. The tryptic peptides were acidified to pH 3 with 10% formic acid and desalted using hand-packed stage tips of C18 resin. The peptides were eluted with 80% acetonitrile and 0.1% formic acid and were dried by vacuum centrifugation. The analysis of tryptic peptides was performed in a NanoLC connected to an Orbitrap Exploris mass spectrometer (Thermo Fisher Scientific), which was used for the analysis of all samples. The peptide separation was carried out using a Proxeon EASY nLC 1200 System (Thermo Fisher Scientific) fitted with a custom-made C18 column (15 cm × 150 μm ID) packed with HxSil C18 3 μm Resin 100 Å (Hamilton). A gradient of water/acetonitrile/0.1% formic acid was employed for chromatography. The samples were injected into the column and run for 180 min at a flow rate of 0.60 μL/min. The peptide separation began with 1% acetonitrile, increasing to 3% in the first 4 min, followed by a linear gradient from 3% to 23% acetonitrile over 86 min, then another increase from 24% to 80% acetonitrile over 35 min, and finally a 35 min wash at 80% acetonitrile, and then decreasing to 1% acetonitrile for 10 min and kept 1% acetonitrile for another 10 min. The eluted peptides were ionized using positive nanoelectrospray ionization (NSI) and directly introduced into the mass spectrometer with an ion source temperature set at 250°C and an ion spray voltage of 2.1 kV. Full-scan MS spectra (m/z 350–2000) were captured in Orbitrap Exploris at a resolution of 120,000 (m/z 400). The automatic gain control was set to 1e6 for full FTMS scans and 5e4 for MS/MS scans. Ions with intensities above 1500 counts underwent fragmentation via NSI in the linear ion trap. The top 15 most intense ions with charge states of ≥2 were sequentially isolated and fragmented using normalized collision energy of 30%, activation Q of 0.250, and an activation time of 10 ms. Ions selected for MS/MS were excluded from further selection for 3 s. The Orbitrap Exploris mass spectrometer was operated using Thermo Xcalibur software.

## Data analysis in MaxQuant

Raw MS files were analyzed in the MaxQuant environment, version 2.4.1.0. The MS/MS spectra were searched using the Andromeda search engine against the UniProt-mouse protein database (UP000000589, December 2021, 55086 entries) and UniProt-*E. coli* protein database (UP000002032, July 2009, 4156 entries). Precursor mass and fragment mass were set with initial mass tolerances of 20 ppm for both the precursor and fragment ions. The search included variable modifications of asparagine/glutamine deamidation, methionine oxidation, and N-terminal acetylation and a fixed modification of carbamidomethyl cysteine. The maximum number of missed cleavages was set at two, and the maximum modifications/peptide and minimum peptide length were set at six amino acids. The UniProt database was also concatenated with an automatically generated reverse database to estimate the false discovery rate (FDR) by using a target decoy search. The FDR was set at 0.01 for the peptide spectrum match and protein identifications. When identified peptides were all shared between two proteins, they were combined and reported as one protein group. For relative quantification, MaxQuant's label-free quantification (LFQ) method was enabled. For *Figure 5—figure supplement 2*, DIA-NN 1.9.1 was used to process the data with enabled contaminants, Enable FASTA digest for library-free search/library generation and deep learning-based spectra, RTs, and IMs prediction using the *M. musculus* reference proteome (UP000000589).

## Statistical analysis

Each treatment and control sample was collected from three technical replicates. The ProteinGroups.txt output file from MaxQuant was exported into Perseus v1.6.15.0 for downstream analysis. In-house functions of Perseus were used to identify and remove protein groups from the reverse decoy database, those marked as potential contaminants, or those only identified by a posttranslational modification site. The remaining intensity and LFQ intensity were $\log_2$-transformed and normalized to the mean Peptidisc peptide intensity value. To identify proteins with significant change in thermal stability, a Student's t-test was conducted with a within-groups variance, s0, set to 0.1. The test was applied to data filtered for proteins that had at least three valid LFQ intensities in either the treatment or control group. The remaining undefined intensity values were imputed from a normal distribution with a downshift of 1.8 standard deviations from the total sample mean and a width of 0.3 times the sample standard deviation. To determine significantly stabilized or destabilized as a result of ligand exposure, proteins with a $-\log_{10}$ p-value >1.3 and LFQ peptide intensity $\log_2$ fold change >1 or <−1 were considered as the direct or indirect candidates of ligand binders. Only proteins with at least two unique peptides were considered for this calculation, a threshold selected because it aligns with established LC–MS/MS data analysis practices (*The and Käll, 2020*; *Kurzawa et al., 2021*; *Reinhard et al., 2015*). A table of the unique peptide count for the discussed proteins in the manuscript is provided in *Supplementary file 3*. To visualize the temperature-dependent peptide intensity data, the smoothed curves were generated using the Piecewise Cubic Hermite Interpolating Polynomial (PCHIP) method. PCHIP interpolation was applied to the mean intensity values of each experimental group across the temperature gradient. Figures were generated through NumPy, Matplotlib, and Pandas Python coding language.

## Ligand-binding prediction with AlphaFold3

The web-based AlphaFold server (https://alphafoldserver.com) was utilized to predict ligand binding onto predicted protein structures. Protein peptide sequences were obtained from UniProt for structure prediction. The resulting output data file was analyzed in Pymol to assess the corresponding B-factor/pLDDT score for each residue interacting with the ligand of interest. The mean pLDDT score from all interacting residues was determined to generate a confidence prediction of ligand binding.

## Protein annotation

The protein list obtained from MaxQuant was subjected to a GO term analysis using the UniProtKB database to identify proteins with the GO term 'membrane'. Proteins from this group were extracted in FastA format, and the Phobius web server (http://phobius.sbc.su.se/) was utilized to predict the number of transmembrane segments (TMS). Any protein with at least one TMS was classified as an IMP. Protein with no TMS but with the GO annotation 'membrane' was classified as a membrane-associated protein; all other proteins were considered as soluble proteins. To assess molecular

functions of significantly stabilized IMPs, the Gene Ontology Molecular Function ('GO_MF_Direct/ GO_MF_FAT') was used through DAVID Bioinformatics (https://david.ncifcrf.gov/). An EASE score of 0.05 was applied to test for significant GO terms based on a p-value cutoff of 0.05 after Benjamini–Hochberg correction. The gProfiler g:GOSt tool was utilized to reduce the redundancy of significantly enriched terms (https://biit.cs.ut.ee/gprofiler/gost).

## Acknowledgements

This work was supported by the Canadian Institutes of Health Research (CIHR; FDN-154318 to MB and PG20R34019 to FD). RSJ holds a CGS-M scholarship from the Natural Sciences & Engineering Research Council of Canada (NSERC). AB holds an FYF scholarship from UBC and an Amplify Doctoral Award from Triangle (training a new generation of researchers in Gastroenterology and liver). MA-S is a recipient of the CBR/LSI Summer Studentship Program.

## Additional information

### Competing interests

Franck Duong van Hoa: Scientific founder of Peptidisc Biotech. The other authors declare that no competing interests exist.

### Funding

| Funder | Grant reference number | Author |
| --- | --- | --- |
| Canadian Institutes of Health Research | FDN-154318 | Mohan Babu |
| Canadian Institutes of Health Research | PG20R34019 | Franck Duong van Hoa |

The funders had no role in study design, data collection and interpretation, or the decision to submit the work for publication.

### Author contributions

Rupinder Singh Jandu, Ashim Bhattacharya, Conceptualization, Data curation, Formal analysis, Validation, Investigation, Visualization, Methodology, Writing – original draft, Writing – review and editing; Frank Antony, Conceptualization, Data curation, Formal analysis, Investigation, Methodology; Mohammed Al-Seragi, Formal analysis, Visualization, Methodology, Writing – original draft; Hiroyuki Aoki, Resources, Data curation, Formal analysis, Methodology; Mohan Babu, Resources, Supervision, Funding acquisition, Project administration, Writing – review and editing; Franck Duong van Hoa, Conceptualization, Resources, Formal analysis, Supervision, Funding acquisition, Project administration, Writing – review and editing

### Author ORCIDs

Rupinder Singh Jandu ⓘ https://orcid.org/0009-0008-9831-2892
Ashim Bhattacharya ⓘ https://orcid.org/0009-0005-8190-9404
Hiroyuki Aoki ⓘ https://orcid.org/0009-0005-9143-086X
Mohan Babu ⓘ https://orcid.org/0000-0003-4118-6406
Franck Duong van Hoa ⓘ https://orcid.org/0000-0001-7328-6124

### Ethics

The C57BL/6 mice were kept in specific pathogen-free conditions and received humane care in compliance with the Canadian Council of Animal Care guidelines, and the animal protocol A23-0280 was approved by the Animal Care Committee of the University of British Columbia.

Reviewer #1 (Public review): https://doi.org/10.7554/eLife.104549.3.sa1
Reviewer #2 (Public review): https://doi.org/10.7554/eLife.104549.3.sa2
Author response https://doi.org/10.7554/eLife.104549.3.sa3

## Additional files

### Supplementary files

Supplementary file 1. Data used to make figures in this manuscript involving volcano plot comparison values or protein intensity values for plots that did not involve volcano plot comparisons.

Supplementary file 2. Total and integral membrane protein (IMP) counts across temperatures in mouse liver peptidisc libraries. The % value is used to assess IMP loss relative to total protein count. Each protein was identified based on at least two unique peptides (n=2).

Supplementary file 3. Unique peptide counts for proteins highlighted in this study: MsbA, ABCA6, ABCB1, ABCB6, ABCC2, ABCC3, ABCC9, ABCG2, ABCG5, BCS1L, P2RX4, P2RY6, P2RY12, and MAO-B. Counts are from MaxQuant analyses using a data-dependent acquisition (DDA) workflow across the temperatures tested in membrane-mimetic thermal proteome profiling (MM-TPP) (with ATP–VO$_4$, 2-MeS-ADP, and AMP-PNP) and in detergent-based thermal proteome profiling (DB-TPP) (with ATP–VO$_4$).

MDAR checklist

### Data availability

The MS-based proteomics data of this study have been deposited to the ProteomeXchange Consortium via the PRIDE partner repository and are available through the identifiers PXD055093 and PXD068828.

The following datasets were generated:

| Author(s) | Year | Dataset title | Dataset URL | Database and Identifier |
| --- | --- | --- | --- | --- |
| Babu M, Aoki H, Jandu RS, Bhattacharya A, Antony F, Al-Seragi M, Duong van Hoa F | 2025 | Membrane mimetic thermal proteome profiling (MM-TPP) towards mapping membrane protein-ligand interaction dynamics | https://www.ebi.ac.uk/pride/archive/projects/PXD055093 | PRIDE, PXD055093 |
| Babu M, Aoki H, Jandu RS, Bhattacharya A, Antony F, Al-Seragi M, Duong van Hoa F | 2025 | Membrane mimetic thermal proteome profiling (MM-TPP) towards mapping membrane protein-ligand interaction dynamics | https://www.ebi.ac.uk/pride/archive/projects/PXD068828 | PRIDE, PXD068828 |

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
