## [Editor Report · eLife Assessment]

This **valuable** study introduces the peptidisc-TPP approach as a promising solution to challenges in membrane proteomics, enabling thermal proteome profiling in a detergent-free system. The concept is innovative and holds significant potential, and the demonstration of its utility and validation is **solid**. The method presents a strong foundation for broader applications in identifying physiologically and pharmacologically relevant membrane protein-ligand interactions.

---

## [Referee Report · Reviewer #1 (Public review)]

Summary:

The idea is appealing, but the authors have not sufficiently demonstrated the utility of this approach.

Strengths:

Novelty of the approach, potential implications for discovering novel interactions

Comments on revisions:

The authors have adequately addressed most of my concerns in this improved version of the manuscript

---

## [Referee Report · Reviewer #2 (Public review)]

Summary:

The membrane mimetic thermal proteome profiling (MM-TPP) presented by Jandu et al. promises a useful way to minimize the interference of detergents in efficient mass spectrometry analysis of membrane proteins. Thermal proteome profiling is a mass spectrometric method that measures binding of a drug to different proteins in a cell lysate by monitoring thermal stabilization of the proteins because of the interaction with the ligands that are being studied. This method has been underexplored for membrane proteome because of the inefficient mass spectrometric detection of membrane proteins and because of the interference from detergents that are used often for membrane protein solubilization.

Strengths:

In this report the binding of ligands to membrane protein targets has been monitored in crude membrane lysates or tissue homogenates exalting the efficacy of the method to detect both intended and off-target binding events in a complex physiologically relevant sample setting. The manuscript is lucidly written and the data presented seems clear. Kudos to the authors. This methodology shows immense potential for identifying membrane protein binders (small-molecule or protein) in a near-native environment, and as a result promises to be a great tool for drug discovery campaigns.

Weaknesses:

While this is a solid report and a promising tool for analyzing membrane protein drug interactions in a detergent-free environment, it is crucial to bear in mind that the process of reconstitution begins with detergent solubilization of the proteome and does not completely circumvent structural perturbations invoked by detergents.

---

## [Author Response]

The following is the authors’ response to the original reviews.

**Reviewer #1 (Public Review):**
Summary:The idea is appealing, but the authors have not sufficiently demonstrated the utility of this approach.Strengths:Novelty of the approach, potential impli=cations for discovering novel interactionsWeaknesses:The Duong had introduced their highly elegant peptidisc approach several years ago. In this present work, they combine it with thermal proteome profiling (TPP) and attempt to demonstrate the utility of this combination for identifying novel membrane protein-ligand interactions.While I find this idea intriguing, and the approach potentially useful, I do not feel that the authors had sufficiently demonstrated the utility of this approach. My main concern is that no novel interactions are identified and validated. For the presentation of any new methodology, I think this is quite necessary. In addition, except for MsbA, no orthogonal methods are used to support the conclusions, and the authors rely entirely on quantifying rather small differences in abundances using either iBAQ or LFQ.

We thank the reviewer for their thoughtful comments. In this revision, we have experimentally addressed the reviewer’s concerns in three ways:

(1) To demonstrate the utility of our MM-TPP method over the detergent-based TPP workflow (termed DB-TPP), we performed a side-by-side comparison using ATP–VO₄ at 51 °C (Figure 3B and Figure 4A). From the DB-TPP dataset, 7.4% of all identified proteins were annotated as ATP-binding, while 6.4% of proteins differentially stabilized were annotated as ATP-binding. In contrast, in the MM-TPP dataset, 9.3% of all identified proteins were annotated as ATP-binding proteins, while 17% of proteins differentially stabilized were annotated as ATP-binding. The lack of enrichment in the detergent-based approach indicates that the observed differences are likely stochastic, rather than a result of specific ATP–VO₄-mediated stabilization as found with MM-TPP. For instance, several key proteins—BCS1, P2RY6, SLC27A2, ABCB1, ABCC2, and ABCC9— found differentially stabilized using the MM-TPP method showed no such pattern in the DB-TPP dataset. This divergence strongly supports the specificity and utility of our Peptidisc approach.

(2) To demonstrate that MM-TPP can resolve not only the broader effects of ATP–VO₄ but also specific ligand–protein interactions, we employed 2-methylthio-ADP (2-MeS-ADP), a selective agonist of the P2RY12 receptor [PMID: 24784220]. In that case, we observed clear thermal stabilization of P2RY12, with more than 6-fold increase in stability at both 51 °C and 57 °C (–log₁₀ p > 5.97; Figure 4B and Figure S4). Notably, no other proteins—including the structurally related but non-responsive P2RY6 receptor- showed comparable stabilization fold change at these temperatures.

(3) To further probe the reproducibility of the method, we performed an independent MMTPP evaluation with ATP–VO₄ at 51 °C using data-independent acquisition (DIA), in contrast to the data-dependent acquisition (DDA) approach used in the initial study (Figure S5). Overall, 7.8% of all identified proteins were annotated as ATP-binding, and as before, this proportion increased to 17% among proteins with log₂ fold changes greater than 0.5. Specifically, BCS1 and SLC27A2 exhibited strong stabilization (log₂ fold change > 1), while P2RY6, ABCB11, ABCC2, and ABCG2 showed moderate stabilization (log₂ fold changes between 0.5 and 1), and consistent with previous results, P2RX4 was destabilized, with a log₂ fold change below –1. These findings support the consistency and reproducibility of the method across distinct data acquisition methods.

My main concern is that no novel interactions are identified and validated. For the presentation of any new methodology, I think this is quite necessary.

The primary objective of our study is to establish and benchmark the MM-TPP workflow using known targets, rather than to discover novel ligand–protein interactions. Identifying new binders requires extensive screening and downstream validations, which we believe is beyond the scope of this methodological report. Instead, our study highlights the sensitivity and reliability of the MM-TPP approach by demonstrating consistent and reproducible results with well-characterized interactions.

We respectfully disagree with the notion that introducing a new methodology must necessarily include the discovery of novel interactions. For instance, Martinez Molina et al. [PMID: 23828940] introduced the cellular thermal shift assay (CETSA) by validating established targets such as MetAP2 with TNP-470 and CDK2 with AZD-5438, without identifying novel protein–ligand pairs. Similarly, Kalxdorf et al. [PMID: 33398190] published their cell-surface thermal proteome profiling (CS-TPP) using Ouabain to stabilize the Na⁺/K⁺-ATPase pump in K562 cells, and SB431542 to stabilize its canonical target JAG1. In fact, when these methods revealed additional stabilizations, these were not validated but instead interpreted through reasoning grounded in the literature. For instance, they attributed the SB431542-induced stabilization of MCT1 to its reported role in cell migration and tumor invasiveness, and explained that SLC1A2 stabilization is related to the disruption of Na⁺/K⁺-ATPase activity by Ouabain. In the same way, our interpretation of ATP-VO₄–mediated stabilization of Mao-B is justified by predictive AlphaFold-3 rather than direct orthogonal assays, which are beyond the scope of our methodological presentation.

Collectively, the influential studies cited above have set methodological precedents by prioritizing validation and proof-of-concept over merely finding uncharacterized binders. In the same spirit, our work is centred on establishing MM-TPP as a robust platform for probing membrane protein–ligand interactions in a water-soluble format. The discovery of novel binders remains an exciting future direction—one that will build upon the methodological foundation laid by the present study.

In addition, except for MsbA, no orthogonal methods are used to support the conclusions, and the authors rely entirely on quantifying rather small differences in abundances using either iBAQ or LFQ.

We deliberately began this study with our model protein, MsbA, examined under both native and overexpressed conditions, to establish an adequation between MMTPP (Figure 2D) and biochemical stability assays (Figure 2A). This validation has provided us with the foundation to confidently extend MM-TPP to the mouse organ proteome. To demonstrate the validity of our workflow, we have used ATP-VO₄ because it has expected targets.

We note that orthogonal validation often requires overproduction and purification of the candidate proteins, including suitable antibodies, which is a true challenge for membrane proteins. Here, we demonstrate that MM-TPP can detect ligand-induced thermal shifts directly in native membrane preparations, without requiring protein overproduction or purification. We also emphasize several influential studies in TPP, including Martinez Molina et al. (PMID: 23828940) and Fang et al. (PMID: 34188175), which focused primarily on establishing and benchmarking the methodology, rather than on extensive orthogonal validation. In the same spirit, our study prioritizes methodological development, and accordingly, several orthogonal validations are now included in this revision.

[...] and the authors rely entirely on quantifying rather small differences in abundances using either iBAQ or LFQ.

To clarify, all analyses on ligand-induced stabilization or destabilization were carried out using LFQ values. The sole exception is on Figure 2B, where we used iBAQ values to depict the relative abundance of proteins within a single sample; this to show MsbA's relative level within the *E. coli* peptidisc library.

Respectfully, we disagree with the assertion that we are “quantifying rather small differences in abundances using either iBAQ or LFQ.” We were able to clearly distinguish between stabilizations driven by specific ligands binding to their targets versus those caused by non-specific ligands with broader activity. This is further confirmed by comparing 2-MeS-ADP, a selective ligand for P2RY12, with ATP-VO₄, a highly promiscuous ligand, and AMP-PNP, which exhibits intermediate breadth. When tested in triplicate at 51 °C, 2-MeS-ADP significantly altered the thermal stability of 27 proteins, AMP-PNP 44 proteins, and ATP-VO₄ 230 proteins, consistent with the expectation that broader ligands stabilize more proteins nonspecifically. Importantly, 2-MeS-ADP produced markedly stronger stabilization of its intended target, P2RY12 (–log_10_p = 9.32), than the top stabilized proteins for ATP–VO₄ (DNAJB3, –log₁₀p = 5.87) or AMP-PNP (FTH1, p = 5.34). Moreover, 2-MeS-ADP did not significantly stabilize proteins that were consistently stabilized by the broad ligands, such as SLC27A2, which was strongly stabilized by both ATP-VO_4_ and AMP-PNP (–log_10_ p>2.5). Together, these findings demonstrate that MMTPP can robustly distinguish between broad-spectrum and target-specific ligands, with selective ligands inducing stronger and more physiologically meaningful stabilization at their intended targets compared to promiscuous ligands.

Finally, we emphasize that our findings are not marginal, but meet quantitative and statistical rigor consistent with best practices in proteomics. We apply dual thresholds combining effect size (|log₂FC| ≥ 1, i.e., at least a two-fold change) with statistical significance (FDR-adjusted p ≤ 0.05)—criteria commonly used in proteomics methodology studies (e.g., PMID: 24942700, 38724498). Moreover, the stabilization and destabilization events we report are reproducible across biological replicates (n = 3), consistent across adjacent temperatures for most targets, and technically robust across acquisition modes (DDA vs. DIA). Taken together, these results reflect statistically valid and biologically meaningful effects, fully aligned with standards set by prior published proteomics studies.

Furthermore, the reported changes in abundances are solely based on iBAQ or LFQ analysis. This must be supported by a more quantitative approach such as SILAC or labeled peptides. In summary, I think this story requires a stronger and broader demonstration of the ability of peptidisc-TPP to identify novel physiologically/pharmacologically relevant interactions.

With respect to labeling strategies, we deliberately avoided using TMT due to concerns about both cost and potential data quality issues. Some recent studies have documented the drawbacks of TMT in contexts directly relevant to our work. For example, a benchmarking study of LiP-MS workflows showed that although TMT increased proteome depth and reduced technical variance, it was less accurate in identifying true drug–protein interactions and produced weaker dose–response correlations compared with label-free DIA approaches [PMID: 40089063]. More broadly, technical reviews have highlighted that isobaric tagging is intrinsically prone to ratio compression and reporterion interference due to co-isolation and co-fragmentation of peptides, which flatten measured fold-changes and obscure biologically meaningful differences [PMID: 22580419, 22036744]. In terms of SILAC, the technique requires metabolic incorporation of heavy amino acids, which is feasible in cultured cells but not in physiologically relevant tissues such as the liver organ used here. SILAC mouse models exist, but they are expensive and time-consuming [PMID: 18662549, 21909926]. We are not a mouse lab, and introducing liver organ SILAC labeling in our workflow is beyond the scope of these revisions. We also note that several hallmark TPP studies have been successfully carried out using label-free quantification [PMID: 25278616, 26379230, 33398190, 23828940], establishing this as an accepted and widely applied approach in the field.

To further support our conclusions, we added controls showing that detergent solubilization of mouse liver membranes followed by SP4 cleanup fails to detect ATP-VO₄– mediated stabilization of ATP-binding proteins, underscoring the necessity of Peptidisc reconstitution for capturing ligand-induced thermal stabilization. We also present new data demonstrating selective stabilization of the P2Y12 receptor by its agonist 2-MeS-ADP, providing orthogonal, receptor-specific validation within the MM-TPP framework. Finally, an orthogonal DIA acquisition on separate replicates confirmed robust ATP-vanadate stabilization of ATP-binding proteins, including BCS1l and SLC27A2. Together, these additions reinforce that the observed stabilizations are genuine, physiologically relevant ligand–protein interactions and highlight the unique advantage of the Peptidisc-based workflow in capturing such events.

Cited Reference:

24784220: Zhang J, Zhang K, Gao ZG, et al. Agonist-bound structure of the human P2Y₁₂ receptor. Nature. 2014;509(7498):119-122. doi:10.1038/nature13288.

23828940: Martinez Molina D, Jafari R, Ignatushchenko M, et al. Monitoring drug target engagement in cells and tissues using the cellular thermal shift assay. Science. 2013;341(6141):84-87. doi:10.1126/science.1233606.

33398190: Kalxdorf M, Günthner I, Becher I, et al. Cell surface thermal proteome profiling tracks perturbations and drug targets on the plasma membrane. Nat Methods. 2021;18(1):84-91. doi:10.1038/s41592-020-01022-1.

34188175: Fang S, Kirk PDW, Bantscheff M, Lilley KS, Crook OM. A Bayesian semi-parametric model for thermal proteome profiling. Commun Biol. 2021;4(1):810. doi:10.1038/s42003-021-02306-8.

24942700: Cox J, Hein MY, Luber CA, Paron I, Nagaraj N, Mann M. Accurate proteome-wide label-free quantification by delayed normalization and maximal peptide ratio extraction, termed MaxLFQ. Mol Cell Proteomics. 2014;13(9):2513-2526. doi:10.1074/mcp.M113.031591.

38724498: Peng H, Wang H, Kong W, Li J, Goh WWB. Optimizing differential expression analysis for proteomics data via high-performing rules and ensemble inference. Nat Commun. 2024;15(1):3922. doi:10.1038/s41467-02447899-w.

40089063: Koudelka T, Bassot C, Piazza I. Benchmarking of quantitative proteomics workflows for limited proteolysis mass spectrometry. Mol Cell Proteomics. 2025;24(4):100945. doi:10.1016/j.mcpro.2025.100945.

22580419: Christoforou AL, Lilley KS. Isobaric tagging approaches in quantitative proteomics: the ups and downs. Anal Bioanal Chem. 2012;404(4):1029-1037. doi:10.1007/s00216-012-6012-9.

22036744: Christoforou AL, Lilley KS. Isobaric tagging approaches in quantitative proteomics: the ups and downs. Anal Bioanal Chem. 2012;404(4):1029-1037. doi:10.1007/s00216-012-6012-9.

18662549: Krüger M, Moser M, Ussar S, et al. SILAC mouse for quantitative proteomics uncovers kindlin-3 as an essential factor for red blood cell function. Cell. 2008;134(2):353-364. doi:10.1016/j.cell.2008.05.033.

21909926: Zanivan S, Krueger M, Mann M. In vivo quantitative proteomics: the SILAC mouse. Methods Mol Biol. 2012;757:435-450. doi:10.1007/978-1-61779-166-6_25.

25278616: Kalxdorf M, Becher I, Savitski MM, et al. Temperature-dependent cellular protein stability enables highprecision proteomics profiling. Nat Methods. 2015;12(12):1147-1150. doi:10.1038/nmeth.3651.

26379230: Savitski MM, Reinhard FBM, Franken H, et al. Tracking cancer drugs in living cells by thermal profiling of the proteome. Science. 2015;346(6205):1255784. doi:10.1126/science.1255784.

33452728: Leuenberger P, Ganscha S, Kahraman A, et al. Cell-wide analysis of protein thermal unfolding reveals determinants of thermostability. Science. 2020;355(6327):eaai7825. doi:10.1126/science.aai7825.

23066101: Savitski MM, Zinn N, Faelth-Savitski M, et al. Quantitative thermal proteome profiling reveals ligand interactions and thermal stability changes in cells. Nat Methods. 2013;10(12):1094-1096. doi:10.1038/nmeth.2766.

30858367: Piazza I, Kochanowski K, Cappelletti V, et al. A machine learning-based chemoproteomic approach to identify drug targets and binding sites in complex proteomes. Nat Commun. 2019;10(1):1216. doi:10.1038/s41467019-09199-0.

**Reviewer #2 (Public Review):**
Summary:The membrane mimetic thermal proteome profiling (MM-TPP) presented by Jandu et al. seems to be a useful way to minimize the interference of detergents in efficient mass spectrometry analysis of membrane proteins. Thermal proteome profiling is a mass spectrometric method that measures binding of a drug to different proteins in a cell lysate by monitoring thermal stabilization of the proteins because of the interaction with the ligands that are being studied. This method has been underexplored for membrane proteome because of the inefficient mass spectrometric detection of membrane proteins and because of the interference from detergents that are used often for membrane protein solubilization.Strengths:In this report the binding of ligands to membrane protein targets has been monitored in crude membrane lysates or tissue homogenates exalting the efficacy of the method to detect both intended and off-target binding events in a complex physiologically relevant sample setting.The manuscript is lucidly written and the data presented seems clear. The only insignificant grammatical error I found was that the 'P' in the word peptidisc is not capitalized in the beginning of the methods section "MM-TPP profiling on membrane proteomes". The clear writing made it easy to understand and evaluate what has been presented. Kudos to the authors.Weaknesses:While this is a solid report and a promising tool for analyzing membrane protein drug interactions, addressing some of the minor caveats listed below could make it much more impactful.The authors claim that MM-TPP is done by "completely circumventing structural perturbations invoked by detergents[1] ". This may not be entirely accurate, because before reconstitution of the membrane proteins in peptidisc, the membrane fractions are solubilized by 1% DDM. The solubilization and following centrifugation step lasts at least for 45 min. It is less likely that all the structural perturbations caused by DDM to various membrane proteins and their transient interactions become completely reversed or rescued by peptidisc reconstitution.

We thank the reviewer for this insightful comment. In response, we have revised the sentence and expanded the discussion to clarify that the Peptidisc provides a complementary approach to detergent-based preparations for studying membrane proteins, preserving native lipid–protein interactions and stabilization effects that may be diminished in detergent.

To further address the structural perturbations invoked by detergents, and as already detailed to our response to Reviewer 1, we have compared the thermal profile of the Peptidisc library to the mouse liver membranes solubilized with 1% DDM, after incubation with ATP–VO₄ at 51 °C (Figure 4A). The results with the detergent extract revealed random patterns of stabilization and destabilization, with only 6.4% of differentially stabilized proteins being ATP-binding—comparable to the 7.4% observed in the background. In contrast, in the Peptidisc library, 17% of differentially stabilized proteins were ATP-binding, compared to 9.3% in the background. Thus, while Peptidisc reconstitution does not fully avoid initial detergent exposure, these findings underscore the importance of implementing Peptidisc in the TPP workflow when dealing with membrane proteins.

In the introduction, the authors make statements such as "..it is widely acknowledged that even mild detergents can disrupt protein structures and activities, leading to challenges in accurately identifying drug targets.." and "[peptidisc] libraries are instrumental in capturing and stabilizing IMPs in their functional states while preserving their interactomes and lipid allosteric modulators...'. These need to be rephrased, as it has been shown by countless studies that even with membrane protein suspended in micelles robust ligand binding assays and binding kinetics have been performed leading to physiologically relevant conclusions and identification of protein-protein and protein-ligand interactions.

We thank the reviewer for this valuable feedback and fully agree with the point raised. In response, we have revised the Introduction and conclusion to moderate the language concerning the limitations of detergent use. We now explicitly acknowledge that numerous studies have successfully used detergent micelles for ligand-binding assays and kinetic analyses, yielding physiologically relevant insights into both protein–protein and protein–ligand interactions [e.g., PMID: 22004748, 26440106, 31776188].

At the same time, we clarify that the Peptidisc method offers a complementary advantage, particularly in the context of thermal proteome profiling (TPP), which involves mass spectrometry workflows that are incompatible with detergents. In this setting, Peptidiscs facilitate the detection of ligand-binding events that may be more difficult to observe in detergent micelles.

We have reframed our discussion accordingly to present Peptidiscs not as a replacement for detergent-based methods, but rather as a complementary tool that broadens the available methodological landscape for studying membrane protein interactions.

If the method involves detergent solubilization, for example using 1% DDM, it is a bit disingenuous to argue that 'interactomes and lipid allosteric modulators' characterized by lowaffinity interactions will remain intact or can be rescued upon detergent removal. Authors should discuss this or at least highlight the primary caveat of the peptidisc method of membrane protein reconstitution - which is that it begins with detergent solubilization of the proteome and does not completely circumvent structural perturbations invoked by detergents.

We would like to clarify that, in our current workflow, ligand incubation occurs after reconstitution into Peptidiscs. As such, the method is designed to circumvent the negative effects of detergent during the critical steps involving low-affinity interactions.

That said, we fully acknowledge that Peptidisc reconstitution begins with detergent solubilization (e.g., 1% DDM), and we have revised the conclusion to explicitly state this important caveat. As the reviewer correctly points out, this initial step may introduce some structural perturbations or result in the loss of weakly associated lipid modulators.

However, reconstitution into Peptidiscs rapidly restores a detergent-free environment for membrane proteins, which has been shown in our previous studies [PMID: 38577106, 38232390, 31736482, 31364989] to mitigate these effects. Specifically, we have demonstrated that time-limited DDM exposure, followed by Peptidisc reconstitution, minimizes membrane protein delipidation, enhances thermal stability, retains functionality, and preserves multi-protein assemblies.

It would also be important to test detergents that are even milder than 1% DDM and ones which are harsher than 1% DDM to show that this method of reconstitution can indeed rescue the perturbations to the structure and interactions of the membrane protein done by detergents during solubilization step.

We selected 1% DDM based on our previous work [PMID: 37295717, 39313981,38232390], where it consistently enabled robust and reproducible solubilization for Peptidisc reconstitution. We agree that comparing milder detergents (e.g., LMNG) and harsher ones (e.g., SDC) would provide valuable insights into how detergent strength influences structural perturbations, and how effectively these can be mitigated by Peptidisc reconstitution. Preliminary data (not shown) from mouse liver membranes indicate broadly similar proteomic profiles following solubilization with DDM, LMNG, and SDC, although potential differences in functional activity or ligand binding remain to be investigated.

Based on the methods provided, it appears that the final amount of detergent in peptidisc membrane protein library was 0.008%, which is ~150 uM. The CMC of DDM depending on the amount of NaCl could be between 120-170 uM.

While we cannot entirely rule out the presence of residual DDM (0.008%) in the raw library, its free concentration may be lower than initially estimated. This is related to the formation of mixed micelles with the amphipathic peptide scaffold, which is supplied in excess during reconstitution. These mixed micelles are subsequently removed during the ultrafiltration step. Furthermore, in related work using His-tagged Peptidiscs [PMID: 32364744], we purified the library by nickel-affinity chromatography following a 5× dilution into a detergent-free buffer. Although this purification step reduced the number of soluble proteins, the same membrane proteins were retained, suggesting that any residual detergent does not significantly interfere with Peptidisc reconstitution. Supporting this, our MM-TPP assays on purified libraries (data not shown) consistently demonstrated stabilization of ATP-binding proteins (e.g., SLC27A2, DNAJB3), indicating that the observed ligand–protein interactions result from successful incorporation into Peptidiscs.

Perhaps, to completely circumvent the perturbations from detergents other methods of detergentfree solubilization such as using SMA polymers and SMALP reconstitution could be explored for a comparison. Moreover, a comparison of the peptidisc reconstitution with detergent-free extraction strategies, such as SMA copolymers, could lend more strength to the presented method.

We agree that detergent-free methods such as SMA polymers hold promise for membrane protein solubilization. However, in preliminary single-replicate experiments using SMA2000 at 51 °C in the presence of ATP–VO₄ (data not shown), we observed broad, non-specific stabilization effects. Of the 2,287 quantified proteins, 9.3% were annotated as ATP-binding, yet 9.9% of the 101 proteins showing a log₂ fold change >1 or <–1 were ATPbinding, indicating no meaningful enrichment. Given this lack of specificity and the limited dataset, we chose not to pursue further SMA experiments and have not included them here. However, in a recent study (https://doi.org/10.1101/2025.08.25.672181), we directly compared Peptidisc, SMA, and nanodiscs for liver membrane proteome profiling. In that work, Peptidisc outperformed both SMA and nanodiscs in detecting membrane protein dysregulation between healthy and diseased liver. By extension, we expect Peptidisc to offer superior sensitivity and specificity for detecting ligand-induced stabilization events, such as those observed here with ATP–vanadate.

Cross-verification of the identified interactions, and subsequent stabilization or destabilizations, should be demonstrated by other in vitro methods of thermal stability and ligand binding analysis using purified protein to support the efficacy of the MM-TPP method. An example cross-verification using SDS-PAGE, of the well-studied MsbA, is shown in Figure 2. In a similar fashion, other discussed targets such as, BCS1L, P2RX4, DgkA, Mao-B, and some un-annotated IMPs shown in supplementary figure 3 that display substantial stabilization or destabilization should be cross-verified.

We appreciate this suggestion and note that a similar point was raised in R1’s comment “In addition, except for MsbA, no orthogonal methods are used to support the conclusions, and the authors rely entirely on quantifying rather small differences in abundances using either iBAQ or LFQ.” We have developed a detailed response to R1 on this matter, which equally applies here.

Cited Reference:

35616533: Young JW, Wason IS, Zhao Z, et al. Development of a Method Combining Peptidiscs and Proteomics to Identify, Stabilize, and Purify a Detergent-Sensitive Membrane Protein Assembly. J Proteome Res. 2022;21(7):1748-1758. doi:10.1021/acs.jproteome.2c00129. PMID: 35616533.

31364989: Carlson ML, Stacey RG, Young JW, et al. Profiling the *Escherichia coli* membrane protein interactome captured in Peptidisc libraries. Elife. 2019;8:e46615. doi:10.7554/eLife.46615.

22004748: O'Malley MA, Helgeson ME, Wagner NJ, Robinson AS. Toward rational design of protein detergent complexes: determinants of mixed micelles that are critical for the in vitro stabilization of a G-protein coupled receptor. Biophys J. 2011;101(8):1938-1948. doi:10.1016/j.bpj.2011.09.018.

26440106: Allison TM, Reading E, Liko I, Baldwin AJ, Laganowsky A, Robinson CV. Quantifying the stabilizing effects of protein-ligand interactions in the gas phase. Nat Commun. 2015;6:8551. doi:10.1038/ncomms9551.

31776188: Beckner RL, Zoubak L, Hines KG, Gawrisch K, Yeliseev AA. Probing thermostability of detergentsolubilized CB2 receptor by parallel G protein-activation and ligand-binding assays. J Biol Chem. 2020;295(1):181190. doi:10.1074/jbc.RA119.010696.

38577106: Jandu RS, Yu H, Zhao Z, Le HT, Kim S, Huan T, Duong van Hoa F. Capture of endogenous lipids in peptidiscs and effect on protein stability and activity. iScience. 2024;27(4):109382. doi:10.1016/j.isci.2024.109382.

38232390: Antony F, Brough Z, Zhao Z, Duong van Hoa F. Capture of the Mouse Organ Membrane Proteome Specificity in Peptidisc Libraries. J Proteome Res. 2024;23(2):857-867. doi:10.1021/acs.jproteome.3c00825.

31736482: Saville JW, Troman LA, Duong Van Hoa F. PeptiQuick, a one-step incorporation of membrane proteins into biotinylated peptidiscs for streamlined protein binding assays. J Vis Exp. 2019;(153). doi:10.3791/60661.

37295717: Zhao Z, Khurana A, Antony F, et al. A Peptidisc-Based Survey of the Plasma Membrane Proteome of a Mammalian Cell. Mol Cell Proteomics. 2023;22(8):100588. doi:10.1016/j.mcpro.2023.100588.

39313981: Antony F, Brough Z, Orangi M, Al-Seragi M, Aoki H, Babu M, Duong van Hoa F. Sensitive Profiling of Mouse Liver Membrane Proteome Dysregulation Following a High-Fat and Alcohol Diet Treatment. Proteomics. 2024;24(23-24):e202300599. doi:10.1002/pmic.202300599.

32364744: Young JW, Wason IS, Zhao Z, Rattray DG, Foster LJ, Duong Van Hoa F. His-Tagged Peptidiscs Enable Affinity Purification of the Membrane Proteome for Downstream Mass Spectrometry Analysis. J Proteome Res. 2020;19(7):2553-2562. doi:10.1021/acs.jproteome.0c00022.

32591519: The M, Käll L. Focus on the spectra that matter by clustering of quantification data in shotgun proteomics. Nat Commun. 2020;11(1):3234. doi:10.1038/s41467-020-17037-3.

33188197: Kurzawa N, Becher I, Sridharan S, et al. A computational method for detection of ligand-binding proteins from dose range thermal proteome profiles. Nat Commun. 2020;11(1):5783. doi:10.1038/s41467-02019529-8.

26524241: Reinhard FBM, Eberhard D, Werner T, et al. Thermal proteome profiling monitors ligand interactions with cellular membrane proteins. Nat Methods. 2015;12(12):1129-1131. doi:10.1038/nmeth.3652.

23828940: Martinez Molina D, Jafari R, Ignatushchenko M, et al. Monitoring drug target engagement in cells and tissues using the cellular thermal shift assay. Science. 2013;341(6141):84-87. doi:10.1126/science.1233606.

32133759: Mateus A, Kurzawa N, Becher I, et al. Thermal proteome profiling for interrogating protein interactions. Mol Syst Biol. 2020;16(3):e9232. doi:10.15252/msb.20199232.

14755328: Dorsam RT, Kunapuli SP. Central role of the P2Y12 receptor in platelet activation. J Clin Invest. 2004;113(3):340-345. doi:10.1172/JCI20986.

**Reviewer #1 (Recommendations for the authors):**
“The authors use iBAC or LFQ to compare across samples. This inconsistency is puzzling. As far as I know, LFQ should always be used when comparing across samples”

As mentioned above, we use iBAQ only in Fig. 2B to illustrate within-sample relative abundance; all comparative analyses elsewhere use LFQ. We have updated the Fig. 2B legend to state this explicitly.

We used iBAQ Fig. 2B as it provides a notion of protein abundance within a sample, normalizing the summed peptide intensities by the number of theoretically observable peptides. This normalization facilitates comparisons between proteins within the same sample, offering a clearer understanding of their relative molar proportions [PMID: 33452728]. LFQ, by contrast, is optimized for comparing the same protein across different samples. It achieves this by performing delayed normalization to reduce run-to-run variability and by applying maximal peptide ratio extraction, which integrates pairwise peptide intensity ratios across all samples to build a consistent protein-level quantification matrix [PMID: 24942700]. These features make LFQ more robust to missing values and technical variation, thereby enabling accurate detection of relative abundance changes in the same protein under different experimental conditions. This distinction is well supported by the proteomics literature: Smits et al. [PMID: 23066101] used iBAQ specifically to determine the relative abundance of proteins within one sample, whereas LFQ was applied for comparative analyses between conditions.

“[Regarding Figure 2A] Why does the control also contain ATP-vanadate? Also, I am not aware of a commercially available chemical "ATP-VO4". I assume this is a mistake”

The control condition in Figure 2A was mislabeled, and the figure has been corrected to remove this discrepancy. In our experiments, ATP and orthovanadate (VO_4_) were added together, and for simplicity this was annotated as “ATP-VO_4_.”

“[Regarding Figure 2B] What is the fold change in MsbA iBAQ values? It seems that the differences are quite small, and as such require a more quantitative approach than iBAQ (e.g SILAC or some other internal standard). In addition, what information does this panel add relative to 2C”

The figure has been updated to clarify that the values shown are log₂transformed iBAQ intensities. Figures 2B and 2C are complementary: Figure 2B shows that in the control sample, MsbA’s peptide abundance decreases with temperatures (51, 56, and 61 °C) relative to the remaining bulk proteins. Figure 2C shows the specific thermal profiles of MsbA in control and ATP–vanadate conditions. To make this clearer, we have added a sentence to the Results section explaining the specific role of Figure 2B.

Together, these panels indicate that the method can identify ligand-induced stabilization even for proteins whose abundance decreases faster than the bulk during the TPP assay. We have provided the rationale for not using SILAC or TMT labeling in our public response.

“[Regarding Figure 2C] Although not mentioned in the legend, I assume this is iBAQ quantification, which as mentioned above isn't accurate enough for such small differences. In addition, I find this data confusing: why is MsbA more stable at the lower temperatures in the absence of ATP-vanadate? The smoothed-line representation is misleading, certainly given the low number of data points”

The data presented represent LFQ values for MsbA, and we have updated the figure legend to clearly indicate this. Additionally, as suggested, we have removed the smoothing line to more accurately reflect the data. Regarding the reviewer’s concern about stability at lower temperatures, we note that MsbA exhibits comparable abundance at 38 °C and 46 °C under both conditions, with overlapping error bars. We therefore interpret these data as indicating no significant difference in stability at the lower temperatures, with ligand-dependent stabilization becoming apparent only at elevated temperatures. We do not exclude the possibility that MsbA stability at these temperatures is affected by the conformational dynamics of this ABC transporter upon ATP binding and hydrolysis.

“[Regarding Figure 3A] is this raw LFQ data? Why did the authors suddenly change from iBAQ to LFQ? I find this inconsistency puzzling”

To clarify, all analyses of protein stabilization or destabilization presented in the manuscript are based on LFQ values. The only instance where iBAQ was used is Figure 2B, where it served to illustrate the relative peptide abundance of MsbA within the same sample. We have revised the figure legends and text to make this distinction explicit and ensure consistency in presentation.

“[Regarding Figure 3B] The non-specific ATP-dependent stabilization increases the likelihood of false positive hits. This limitation is not mentioned by the authors. I think it is important to show other small molecules, in addition to ATP. The authors suggest that their approach is highly relevant for drug screening. Therefore, a good choice is to test an effect of a known stabilizing drug (eg VX-809 and CFTR)”

We thank the reviewer for this suggestion. As noted in the manuscript (results and discussion sections), ATP is a natural hydrotrope and is therefore expected to induce broad, non-specific stabilization effects, a phenomenon also observed in previous proteome-wide studies, which demonstrated ATP’s widespread influence on cytosolic protein solubility and thermal stability (PMID: 30858367). To demonstrate that MM-TPP can resolve specific ligand–protein interactions beyond these global ATP effects, we tested 2-methylthio-ADP (2-MeS-ADP), a selective agonist of P2RY12 (PMID: 14755328). In these experiments, we observed robust and reproducible stabilization of P2RY12 at both 51°C and 57°C, with no consistent stabilization of unrelated proteins across temperatures. This provides direct evidence that our workflow can distinguish specific from non-specific ligand-induced effects. We selected 2-MeS-ADP due to its structural stability and receptor higher-affinity over ADP, allowing us to extend our existing workflow while testing a receptor-specific interaction. We agree that extending this approach to clinically relevant small-molecule drugs, such as VX-809 with CFTR, would further underscore the pharmacological potential of MM-TPP, and we have now noted this as an important avenue for future studies.

“X axis of Figure 3B: Log 2 fold difference of what? iBAQ? LFQ? Similar ambiguity regarding the Y axis of 3E. What peptide? And why the constant changes in estimating abundances?”

We thank the reviewer for pointing out these inaccuracies in the figure annotations. As mentioned above, all analyses (except Figure 2B) are based on LFQ values. We have revised the figure legends and text to make this clear.

In Figure 3E, “peptide intensity” refers to log2 LFQ peptide intensities derived from the BCS1L protein, as indicated in the figure caption.

“The authors suggest that P2RY6 and P2RY12 are stabilized by ADP, the hydrolysis product of ATP. Currently, the support for this suggestion is highly indirect. To support this claim, the authors need to directly show the effect of ADP. In reference to the alpha fold results shown in Figure 4D, the authors state that "Collectively, these data highlight the ability of MM-TPP to detect the side effects of parent compounds, an important consideration for drug development". To support this claim, it is necessary to show that Mao-B is indeed best stabilized with ADP or AMP, rather than ATP.”

In this revision, we chose not to test ADP directly, as it is a broadly binding, relatively weak ligand that would likely stabilize many proteins without revealing clear target-specific effects. Since we had already evaluated ATP-VO₄, a similarly broad, non-specific ligand, additional testing with ADP would provide limited additional insight. Instead, we prioritized 2-methylthio-ADP, a selective agonist of P2RY12, to more effectively demonstrate the specificity of MM-TPP. With this ligand, we observed clear and reproducible stabilization of P2RY12, underscoring the ability of MM-TPP to resolve receptor–ligand interactions beyond ATP’s broad hydrotropic effects. Importantly, and as expected, we did not observe stabilization of the related purinergic receptor P2RY6, further supporting the specificity of the observed effect.

We have also revised the AlphaFold-related statement in Figure 4D to adopt a more cautious tone: “Collectively, these data suggest that MM-TPP may detect potential side effects of parent compounds, an important consideration for drug development.” In this context, we use AlphaFold not as a validation tool, but rather as a structural aid to help rationalize why certain off-target proteins (e.g., ATP with Mao-B) exhibit stabilization.

**Reviewer #2 (Recommendations for the authors):**
“In the main text, it will be useful to include the unique peptides table of at least the targets discussed in the manuscript. For example, in presence of AMP-PNP at 51oC P2RY6 shows 4-6 peptides in all n=3 positive & negative ionization modes. But, for P2RY12 only 1-3 peptides were observed. Depending on the sequence length and the relative abundance in the cell of a protein of interest, the number of peptides observed could vary a lot per protein. Given the unique peptide abundance reported in the supplementary file, for various proteins in different conditions, it appears the threshold of observation of two unique peptides for a protein to be analyzed seems less stringent.”

By applying a filter requiring at least two unique peptides in at least one replicate, we exclude, on average, 15–20% of the total identified proteins. We consider this a reasonable level of stringency that balances confidence in protein identification with the retention of relevant data. This threshold was selected because it aligns with established LC-MS/MS data analysis practices (PMID: 32591519, 33188197, 26524241), and we have included these references in the Methods section to justify our approach. We have included in this revision a Supplemental Table 2 showing the unique peptide counts for proteins highlighted in this study.

“It appears that the time of heat treatment for peptidisc library subjected to MM-TPP profiling was chosen as 3 min based on the results presented in Supplementary Figure 1A, especially the loss of MsbA observed in 1% DDM after 3 min heat perturbation. However, when reconstituted in peptidisc there seems to be no loss in MsbA even after 12 mins at 45oC. So, perhaps a longer heat treatment would be a more efficient perturbation.”

Previous studies indicate that heat exposure of 3–5 minutes is optimal for visualizing protein denaturation (PMID: 23828940, 32133759). We have added a statement to the Results section to justify our choice of heat exposure. Although MsbA remains stable at 45 °C for extended periods, higher temperatures allow for more effective perturbation to reveal destabilization. Supplementary Figure 1A specifically illustrates MsbA instability in detergent environments.

“Some of the stabilized temperatures listed in Table 1 are a bit confusing. For example, ABCC3 and ABCG2. In the case of ABCC3 stabilization was observed at 51oC and 60oC, but 56oC is not mentioned. In the same way, 51oC is not mentioned for ABCG2. You would expect protein to be stabilized at 56oC if it is stabilized at both 51oC and 60oC. So, it is unclear if the stabilizations were not monitored for these proteins at the missing temperatures in the table or if no peptides could be recorded at these temperatures as in the case of P2RX4 at 60oC in Figure 4C.”

Both scenarios are represented in our data. For some proteins, like ABCG2, sufficient peptide coverage was achieved, but no stabilization was observed at intermediate temperatures (e.g., 56 °C), likely because the perturbation was not strong enough to reveal an effect. In other cases, such as ABCC3 at 56 °C or P2RX4 at 60 °C, the proteins were not detected due to insufficient peptide identifications at those temperatures, which explains their omission from the table.

“In Figure 4C, it is perplexing to note that despite n = 3 there were no peptide fragments detected for P2RX4 at 60oC in presence of ATP-VO4, but they were detected in presence of AMP-PNP. It will be useful to learn authors explanation for this, especially because both of these ligands destabilize P2RX4. In Figure 4B, it would have been great to see the effect of ADP too, to corroborate the theory that ATP metabolites could impact the thermal stability.”

In Figure 4C, the absence of P2RX4 peptide detection at 60 °C with ATP–VO₄ mirrors variability observed in the corresponding control (n = 6). Specifically, neither the control nor ATP–VO₄ produced unique peptides for P2RX4 at 60 °C in that replicate, whereas peptides were detected at 60 °C in other replicates for both the control and AMPPNP, and at 64 °C for ATP–VO_4_, the controls, and AMP-PNP. Such missing values are a natural feature of MS-based proteomics and can arise from multiple technical factors, including inconsistent heating, incomplete digestion, stochastic MS injection, or interference from Peptidisc peptides. We therefore interpret the absence of peptides in this replicate as a technical artifact rather than evidence against protein destabilization. Importantly, the overall dataset consistently shows that both ATP–VO₄ and AMP-PNP destabilize P2RX4, supporting their characterization as broad, non-specific ligands with off-target effects.

Because ATP and ADP belong to the same class of broadly binding, non-specific ligands, additional testing with ADP would not provide meaningful mechanistic insight. Instead, we chose to test 2-methylthio-ADP, a selective P2RY12 agonist. This experiment revealed robust, reproducible stabilization of P2RY12, without consistent effects on unrelated proteins at 51 °C and 57 °C, thereby demonstrating the ability of MM-TPP to detect specific receptor–ligand interactions.

Finally, we note that P2RX4 is not a primary target of ATP–VO_4_ or AMP-PNP. Consequently, the observed destabilization of P2RX4 is expected to be less pronounced than the strong, physiologically consistent stabilization of ABC transporters by ATP–VO_4_, as shown in Figure 3D, where the majority of ABC transporters are thermally stabilized across all tested temperatures.

“As per Figure 4, P2Y receptors P2RY6 and P2RY12 both showed great thermal stability in presence of ATP-VO4 despite their preference for ADP. The authors argue this could be because of ATP metabolism, and binding of the resultant ADP to the P2RY6. If P2RX4 prefers ATP and not the metabolized product ADP that apparently is available, ideally you should not see a change in stability. A stark destabilization would indicate interaction of some sorts. P2X receptors are activated by ATP and are not naturally activated by AMP-PNP. So, destabilization of P2RX4 upon binding to ATP that can activate P2X receptors is conceivable. However, destabilization both in presence of ATP-VO4 and AMP-PNP is unclear. It is perhaps useful to test effect of ADP using this method, and maybe even compare some antagonists such as TNPATP.”

In this study, we did not directly test ADP, as we had already demonstrated that MM-TPP detects stabilization by broad-binding ligands such as ATP–VO₄. Instead, we focused on a more selective ligand, 2-MeS-ADP, a specific agonist of P2RY12 [PMID: 14755328]. Here, we observed robust and reproducible stabilization of P2RY12 at 51 °C and 57 °C, while P2RY6 showed no significant changes, and no other proteins were consistently stabilized (Figure 4B, S4). This confirms that MM-TPP can distinguish specific ligand–receptor interactions from broader ATP-induced effects. To further explore the assay’s nuance and sensitivity, testing additional nucleotide ligands—including antagonists like TNP-ATP or ATPγS—would provide valuable insights, and we have identified this as an important future direction.